# GROUNDED OBJECT-CENTRIC LEARNING

**Avinash Kori** [†], **Francesco Locatello** [‡] [*], **Fabio De Sousa Ribeiro**[†],
**Francesca Toni** [†], and **Ben Glocker** [†]
[†] Imperial College London, [‡] Institute of Science and Technology Austria
a.kori21@imperial.ac.uk

## ABSTRACT

The extraction of modular object-centric representations for downstream tasks is an emerging area of research. Learning grounded representations of objects that are guaranteed to be stable and invariant promises robust performance across different tasks and environments. Slot Attention (SA) learns object-centric representations by assigning objects to *slots*, but presupposes a *single* distribution from which all slots are randomly initialised. This results in an inability to learn *specialized* slots which bind to specific object types and remain invariant to identity-preserving changes in object appearance. To address this, we present C*onditional* Sl*ot* A*ttention* (CoSA) using a novel concept of *Grounded Slot Dictionary* (GSD) inspired by vector quantization. Our proposed GSD comprises (i) canonical object-level property vectors and (ii) parametric Gaussian distributions, which define a prior over the slots. We demonstrate the benefits of our method in multiple downstream tasks such as scene generation, composition, and task adaptation, whilst remaining competitive with SA in popular object discovery benchmarks.

## 1 INTRODUCTION

A key step in aligning AI systems with humans amounts to imbuing them with notions of *objectness* akin to human understanding (Lake et al., 2017). It has been argued that humans understand their environment by subconsciously segregating percepts into object entities (Rock, 1973; Hinton, 1979; Kulkarni et al., 2015; Behrens et al., 2018). Objectness is a multifaceted property that can be characterised as physical, abstract, semantic, geometric, or via spaces and boundaries (Yuille & Kersten, 2006; Epstein et al., 2017). The goal of *object-centric representation learning* is to equip systems with a notion of objectness which remains stable, *grounded* and invariant to different environments, such that the learned representations are disentangled and useful (Bengio et al., 2013).

The **grounding problem** refers to the challenge of connecting such representations to real-world objects, their function, and meaning (Harnad, 1990). It can be understood as the challenge of learning abstract, *canonical* representations of objects.[1] The **binding problem** refers to the challenge of how objects are combined into a single context (Revonsuo & Newman, 1999). Both of these problems affect a system's ability to understand the world in terms of symbol-like entities, or true factors of variation, which are crucial for systematic generalization (Bengio et al., 2013; Greff et al., 2020).

Greff et al. (2020) proposed a functional division of the binding problem into three concrete sub-tasks: (i) *segregation*: the process of forming grounded, modular object representations from raw input data; (ii) *representation*: the ability to represent multiple object representations in a common format, without interference between them; (iii) *compositionality*: the capability to dynamically relate and compose object representations without sacrificing the integrity of the constituents. The mechanism of *attention* is believed to be a crucial component in determining which objects appear to be bound together, segregated, and recalled (Vroomen & Keetels, 2010; Hinton, 2022). Substantial progress in object-centric learning has been made recently, particularly in unsupervised objective discovery, using iterative attention mechanisms like Slot Attention (SA) (Locatello et al., 2020) and many others (Engelcke et al., 2019; 2021; Singh et al., 2021; Chang et al., 2022; Seitzer et al., 2022).

---

[*]Work done when the author was a part of AWS.

[1]Representations learned by neural networks are directly grounded in their input data, unlike classical notions of *symbols* whose definitions are often subject to human interpretation.

Despite recent progress, several major challenges in the field of object-centric representation learning remain, including but not limited to: (i) respecting object symmetry and independence, which requires isolating individual objects irrespective of their orientation, viewpoint and overlap with other objects; (ii) dynamically estimating the total number of unique and repeated objects in a given scene;

(iii) the binding of *dynamic* representations of an object with more *permanent* (canonical) identifying characteristics of its *type*. As Treisman (1999) explains, addressing the challenge (iii) is a pre-requisite for human-like perceptual binding. In this work, we argue this view of binding *temporary* object representations (so-called "object files") to their respective *permanent* object *types*, is a matter of learning a vocabulary of *grounded*, canonical object representations. Therefore, our primary focus is on addressing challenge (iii), which we then leverage to help tackle challenges (i) and (ii). To that end, our proposed approach goes beyond standard slot attention (SA) (Locatello et al., 2020), as shown in Figure 1 and described next.

**Contributions.** Slot attention learns composable representations using a dynamic inference-level binding scheme for assigning objects to *slots*, but presupposes a *single* distribution from which all slots are randomly initialised. This results in an inability to learn *specialized* slots that bind to specific object types and remain invariant to identity-preserving changes in object appearance. To address this, we present CO*nditional* S*lot* A*ttention* (CoSA) using a novel *Grounded Slot Dictionary* (GSD) inspired by vector quantization techniques. Our GSD is *grounded* in the input data and consists of: (i) canonical object-level property vectors which are learnable in an

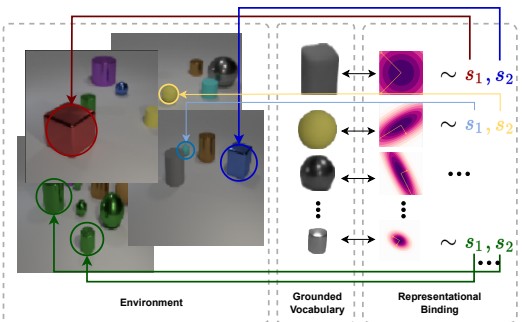

Figure 1: The leftmost block illustrates various scenes within an environment, each featuring different object instances. In the middle block, we depict our acquired *grounded vocabulary* of canonical object-centric representations, effectively capturing object *types*. The rightmost block displays a collection of *specialized slot* distributions associated with their respective canonical representations. These distributions are employed to sample initial slots for object instances within a scene. This process, known as *object binding*, is elucidated by the placeholder slots $s_1$ and $s_2$. These slots are linked to specific object types in the environment and undergo further refinement. Notably, this differs from the SA, which relies on a *single* distribution for random slot initialization and does not encourage slots to remain invariant in the face of identity-preserving changes in object appearance.

unsupervised fashion; and (ii) parametric Gaussian distributions defining a prior over object slots. In summary, our main contributions are as follows:

(i) We propose the *Grounded Slot Dictionary* for object-centric representation learning, which unlocks the capability of learning *specialized* slots that bind to specific object *types*;

(ii) We provide a probabilistic perspective on unsupervised object-centric dictionary learning and derive a principled end-to-end objective for this model class using variational inference methods;

(iii) We introduce a simple strategy for dynamically quantifying the number of unique and repeated objects in a given input scene by leveraging spectral decomposition techniques;

(iv) Our experiments demonstrate the benefits of grounded slot representations in multiple tasks such as scene generation, composition and task adaptation whilst remaining competitive with standard slot attention in object discovery-based tasks.

## 2 RELATED WORK

**Object Discovery.** Several notable works in the field object discovery, including Burgess et al. (2019); Greff et al. (2019); Engelcke et al. (2019), employ an iterative variational inference approach (Marino et al., 2018), whereas Van Steenkiste et al. (2020); Lin et al. (2020) adopt more of a generative perspective. More recently, the use of iterative attention mechanisms in object *slot*-based models has garnered significant interest (Locatello et al., 2020; Engelcke et al., 2021; Singh et al., 2021; Wang et al., 2023; Singh et al., 2022; Emami et al., 2022). The focus of most of these approaches remains centered around the disentanglement of slot representations, which can be understood as tackling the *segregation* and *representation* aspects of the binding problem (Greff et al., 2020). Van Steenkiste

et al. (2020); Lin et al. (2020); Singh et al. (2021) focus primarily on tackling the *composition* aspect of the binding problem, either in a controlled setting or with a predefined set of prompts. However, unlike our approach, they do not specifically learn *grounded* symbol-like entities for scene composition. Another line of research tackling the binding problem involves capsules (Sabour et al., 2017; Hinton et al., 2018; Ribeiro et al., 2020). However, these methods face scalability issues and are typically used for discriminative learning. Kipf et al. (2021); Elsayed et al. (2022) perform conditional slot attention with weak supervision, such as using the center of mass of objects in the scene or object bounding boxes from the first frame. In contrast, our approach learns specialized slot representations fully unsupervised. Our proposed method builds primarily upon slot attention (Locatello et al., 2020) by introducing a *grounded slot dictionary* to learn specialized slots that bind to specific object types in an unsupervised fashion.

**Discrete Representation Learning.** Van Den Oord et al. (2017) propose Vector Quantized Variational Autoencoders (VQ-VAE) to learn discrete latent representations by mapping a continuous latent space to a fixed number of *codebook embeddings*. The codebook embeddings are learned by minimizing the mean squared error between continuous and nearest codebook embeddings. The learning process can also be improved upon using the Gumbel softmax trick Jang et al. (2016); Maddison et al. (2016). The VQ-VAE has been further explored for text-to-image generation (Esser et al., 2021; Gu et al., 2021; Ding et al., 2021; Ramesh et al., 2021). One major challenge in the quantization approach is effectively utilizing codebook vectors. Yu et al. (2021); Santhirasekaram et al. (2022b;a) address this by projecting the codebook vectors onto Hyperspherical and Poincare manifold. Träuble et al. (2022) propose a discrete key-value bottleneck layer and show the effect of discretization on non-i.i.d samples, generalizability, and robustness tasks. We take inspiration from their approach in developing our grounded slot dictionary.

**Compositional Visual Reasoning.** Using grounded object-like representations for compositionality is said to be fundamental for realizing human-level generalization (Greff et al., 2020). Several notable works propose data-driven approaches to first learn object-centric representations, and then use symbol-based reasoning wrappers on top (Garcez et al., 2002; 2019; Hudson & Manning, 2019; Vedantam et al., 2019). Mao et al. (2019) introduced an external, learnable reasoning block for extracting symbolic rules for predictive tasks. Yi et al. (2018); Stammer et al. (2021) use visual question-answering (VQA) for disentangling object-level representations for downstream reasoning tasks. Stammer et al. (2021) in particular also base their approach on a slot attention module to learn object-centric representations, which are then further used for set predictions and rule extraction. Unlike most of these methods, which use some form of dense supervision either in the form of object information or natural language question answers, in this work, we train the model for discriminative tasks and use the emerging properties learned as a result of slot dictionary and iterative refinement for learning rules for the classification.

## 3 BACKGROUND

**Discrete Representation Learning.** Let $\mathbf{z} = \Phi_e(\mathbf{x}) \in \mathbb{R}^{N \times d_z}$ denote an encoded representation of an input datapoint $\mathbf{x}$ consisting of $N$ embedding vectors in $\mathbb{R}^{d_z}$. Discrete representation learning aims to map each $\mathbf{z}$ to a set of elements in a codebook $\mathfrak{S}$ consisting of $M$ embedding vectors. The codebook vectors are randomly initialized at the start of training and are updated to align with $\mathbf{z}$. The codebook is initialised in a specific range based on the choice of sampling as detailed below:

(i) **Euclidean:** The embeddings are randomly initialized between $(-1/M, 1/M)$ as described by Van Den Oord et al. (2017). The discrete embeddings $\tilde{\mathbf{z}}$ are then sampled with respect to the Euclidean distance as follows: $\tilde{\mathbf{z}} = \arg\min_{\mathfrak{S}_j} ||\mathbf{z} - \mathfrak{S}_j||_2^2, \forall \mathfrak{S}_j \in \mathfrak{S}$.

(ii) **Cosine:** The embeddings are initialized on a unit hypersphere (Yu et al., 2021; Santhirasekaram et al., 2022b). The representations $\mathbf{z}$ are first normalized to have unit norm: $\hat{\mathbf{z}} = \mathbf{z}/||\mathbf{z}||$. The discrete embeddings are sampled following: $\tilde{\mathbf{z}} = \arg\max_{\mathfrak{S}_j} \langle \hat{\mathbf{z}}, \mathfrak{S}_j \rangle, \forall \mathfrak{S}_j \in \mathfrak{S}$, where $\langle ., . \rangle$ denotes vector inner product. The resulting discrete representations are then upscaled as $\tilde{\mathbf{z}} \cdot ||\mathbf{z}||$.

(iii) **Gumbel:** The embeddings are initialised similarly to the Euclidian codebook. The representations $\mathbf{z}$ are projected onto discrete embeddings in $\mathfrak{S}$ by measuring the pair-wise similarity between the $\mathbf{z}$ and the $M$ codebook elements. The projected vector $\hat{\mathbf{z}}$ is used in the Gumbel-Softmax trick (Maddison et al., 2016; Jang et al., 2016) resulting in the continuous

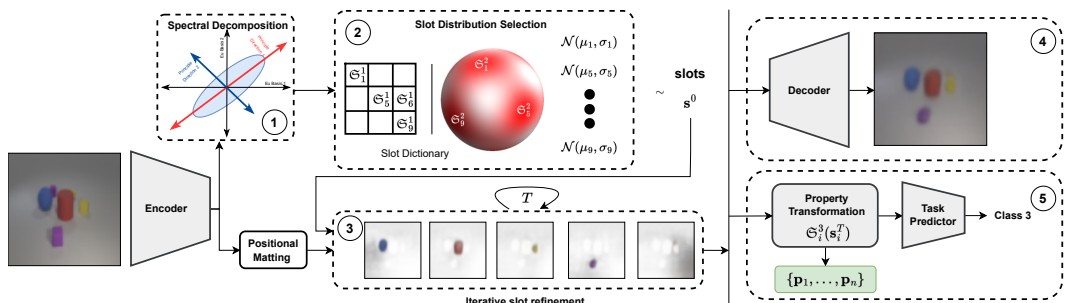

Figure 2: CoSA is an unsupervised autoencoder framework for *grounded* object-centric representation learning, and it is composed of five unique sub-modules. ① The **abstraction** module extracts all the *distinct* objects in a scene using spectral decomposition. ② The **grounded slot dictionary** (GSD) module maps the object representation to *grounded* (canonical) slot representations, which are then used for sampling initial slot conditions. ③ The **refinement** module uses slot attention to iteratively refine the initial slot representations. ④ The **discovery** module maps the slot representations to observational space (used for object discovery and visual scene composition). ⑤ The **reasoning** module involves object property transformation and the prediction model (used for reasoning tasks).

approximation of one-hot representation given by $\mathrm{softmax}((\mathbf{g}_i + \hat{\mathbf{z}}_i)/\tau)$, where $\mathrm{g}_i$ is sampled from a Gumbel distribution, and $\tau$ is a temperature parameter.

**Slot Attention.** Slot attention (Locatello et al., 2020) takes a set of feature embeddings $\mathbf{z} \in \mathbb{R}^{N \times d_z}$ as input, and applies an iterative attention mechanism to produce $K$ object-centric representations called slots $\mathbf{s} \in \mathbb{R}^{K \times d_s}$. Let $\mathcal{Q}_\gamma, \mathcal{K}_\beta$ and $\mathcal{V}_\phi$ denote query, key and value projection networks with parameters $\beta, \gamma$ and $\psi$ respectively acting on $\mathbf{z}$. To simplify our exposition later on, let $f$ and $g$ be shorthand notation for the *slot update* and *attention* functions respectively, defined as:

$$f(\boldsymbol{A}, \mathbf{v}) = \boldsymbol{A}^T \mathbf{v}, \quad A_{ij} = \frac{g(\mathbf{q}, \mathbf{k})_{ij}}{\sum_{l=1}^{K} g(\mathbf{q}, \mathbf{k})_{lj}} \quad \text{and} \quad g(\mathbf{q}, \mathbf{k}) = \frac{e^{M_{ij}}}{\sum_{l=1}^{N} e^{M_{il}}}, \quad \boldsymbol{M} = \frac{\mathbf{k}\mathbf{q}^T}{\sqrt{d_s}}, \quad (1)$$

where $\mathbf{q} = \mathcal{Q}_\gamma(\mathbf{z}) \in \mathbb{R}^{K \times d_s}$, $\mathbf{k} = \mathcal{K}_\beta(\mathbf{z}) \in \mathbb{R}^{N \times d_s}$, and $\mathbf{v} = \mathcal{V}_\phi(\mathbf{z}) \in \mathbb{R}^{N \times d_s}$ are the query, key and value vectors respectively. The attention matrix is denoted by $\boldsymbol{A} \in \mathbb{R}^{N \times K}$. Unlike self-attention (Vaswani et al., 2017), the queries in slot attention are a function of the slots $\mathbf{s} \sim \mathcal{N}(\mathbf{s}; \boldsymbol{\mu}, \boldsymbol{\sigma}) \in \mathbb{R}^{K \times d_s}$, and are iteratively refined over $T$ attention iterations (see refinement module in Fig. 2). The slots are randomly initialized at $t = 0$. The queries at iteration $t$ are given by $\hat{\mathbf{q}}^t = \mathcal{Q}_\gamma(\mathbf{s}^t)$, and the slot update process can be summarized as: $\mathbf{s}^{t+1} = f(g(\hat{\mathbf{q}}^t, \mathbf{k}), \mathbf{v})$. Lastly, a Gated Recurrent Unit (GRU), which we denote by $\mathcal{H}_\theta$, is applied to the slot representations $\mathbf{s}^{t+1}$ at the end of each SA iteration, followed by a generic MLP skip connection.

## 4   UNSUPERVISED CONDITIONAL SLOT ATTENTION: FORMALISM

In this section, we present our proposed conditional slot attention (CoSA) framework using a novel *grounded* slot dictionary (GSD) inspired by vector quantization. CoSA is an unsupervised autoencoder framework for *grounded* object-centric representation learning, and it consists of five sub-modules in total (as depicted in Fig. 2), each of which we will describe in detail next.

**Notation.** Let $\mathcal{D} \subseteq \mathcal{X} \times \mathcal{Y}$ denote a dataset of images $\mathcal{X} \in \mathbb{R}^{H \times W \times C}$ and their *properties* $\mathcal{Y} \subseteq \mathcal{Y}_1 \times \mathcal{Y}_2 \in \mathbb{R}^Y$. There are $Y$ *properties* in total, where $\mathcal{Y}_1$ corresponds to the space of image labels, and $\mathcal{Y}_2$ consists of additional information about the images like object size, shape, location, object material, etc. Let $\Phi_e : \mathcal{X} \to \mathcal{Z}$ denote an encoder and $\Phi_d : \mathcal{Z} \to \mathcal{X}$ denote a decoder, mapping to and from a latent space $\mathcal{Z}$. Further, let $\Phi_r : \mathcal{Z} \to \mathcal{Y}$ denote a classifier for *reasoning* tasks.

① **Abstraction Module.** The purpose of the abstraction module is to enable dynamic estimation of the number of slots required to represent the input. Since multiple instances of the same object can appear in a given scene, we introduce the concept of an *abstraction function* denoted as: $\mathcal{A} : \mathbb{R}^{N \times d_z} \to \mathbb{R}^{\tilde{N} \times d_z}$, which maps $N$ input embeddings to $\tilde{N}$ output embeddings representing $\tilde{N}$ *distinct* objects. This mapping learns the canonical vocabulary of unique objects present in an

environment, as shown in Fig. 1. To accomplish this, we first compute the latent covariance matrix $\boldsymbol{C} = \mathbf{z} \cdot \mathbf{z}^T \in \mathbb{R}^{N \times N}$, where $\mathbf{z} = \Phi_e(\mathbf{x}) \in \mathbb{R}^{N \times d_z}$ are feature representations of an input $\mathbf{x}$. We then perform a spectral decomposition resulting in $\mathbf{C} = \boldsymbol{V} \Lambda \boldsymbol{V}^T$, where $\boldsymbol{V}$ and $\mathrm{diag}(\Lambda)$ are the eigenvector and eigenvalue matrices, respectively. The eigenvectors in $\boldsymbol{V}$ correspond to the directions of maximum variation in $\mathbf{C}$, ordered according to the eigenvalues, which represent the respective magnitudes of variation. We project $\mathbf{z}$ onto the top $\tilde{N}$ principal vectors, resulting in abstracted, property-level representation vectors in a new coordinate system (principal components).

The spectral decomposition yields high eigenvalues when: (i) a single uniquely represented object spans multiple input embeddings in $\mathbf{z}$ (i.e. a large object is present in $\mathbf{x}$); (ii) a scene contains multiple instances of the same object. To accommodate both scenarios, we assume that the maximum area spanned by an object is represented by the highest eigenvalue $\lambda_s$ (excluding the eigenvalue representing the background). Under this assumption, we can dynamically estimate the number of slots required to represent the input whilst preserving maximal explained variance. To that end, we first filter out small eigenvalues by flooring them $\lambda_i = \lfloor \lambda_i \rfloor$, then compute a sum of eigenvalue ratios w.r.t. $\lambda_s$ resulting in a total number of slots required: $K = 1 + \sum_{i=2}^{\tilde{N}} \lceil \lambda_i / \lambda_s \rceil$. Intuitively, this ratio dynamically estimates the number of object instances, relative to the 'largest' object in the input. Note that we do not apply positional embedding to the latent features at this stage, as it encourages multiple instances of the same object to be uniquely represented, which we would like to avoid.

**(2) GSD Module.** To obtain grounded object-centric representations that connect scenes with their fundamental building blocks (i.e. object *types* (Treisman, 1999)), we introduce GSD, as defined in 1.

**Definition 1.** (Grounded Slot Dictionary) A grounded slot dictionary $\mathfrak{S}$ consists of: (i) an object-centric representation codebook $\mathfrak{S}^1$; (ii) a set of slot prior distributions $\mathfrak{S}^2$. The number of objects $M$ in the environment is predefined, and the GSD $\mathfrak{S}$ is constructed as follows:

$$\mathfrak{S} := \left\{ (\mathfrak{S}_i^1, \mathfrak{S}_i^2) \right\}_{i=1}^M, \qquad \tilde{\mathbf{z}} = \underset{\mathfrak{S}_i^1 \in \mathfrak{S}^1}{\arg\min}\, \hat{d}\left( \mathcal{A}(\mathbf{z}), \mathfrak{S}_i^1 \right), \qquad \mathfrak{S}_i^2 = \mathcal{N}\left( \mathbf{s}_i^0; \boldsymbol{\mu}_i, \boldsymbol{\sigma}_i^2 \right), \qquad (2)$$

where $\tilde{\mathbf{z}} \in \mathbb{R}^{\tilde{N} \times d_z}$ denotes the $\tilde{N}$ codebook vector representations closest to the output of the abstraction function $\mathcal{A}(\mathbf{z}) \in \mathbb{R}^{\tilde{N} \times d_z}$, applied to the input encoding $\mathbf{z} = \Phi_e(\mathbf{x}) \in \mathbb{R}^{N \times d_z}$.

As per Equation 2, the codebook $\mathfrak{S}^1$ induces a mapping from input-dependent continuous representations $\mathbf{z}' = \mathcal{A}(\Phi_e(\mathbf{x}))$, to a set of discrete (canonical) object-centric representations $\tilde{\mathbf{z}}$ via a distance function $\hat{d}(\cdot, \cdot)$ (judicious choices for $\hat{d}(\cdot, \cdot)$ are discussed in Section 3). Each slot prior distribution $p(\mathbf{s}_i^0) = \mathcal{N}\left( \mathbf{s}_i^0; \boldsymbol{\mu}_i, \boldsymbol{\sigma}_i^2 \right)$ in $\mathfrak{S}^2$ is associated with one of the $M$ object representations in the codebook $\mathfrak{S}^1$. These priors define marginal distributions over the initial slot representations $\mathbf{s}^0$. We use diagonal covariances and the learn the parameters $\{\boldsymbol{\mu}_i, \boldsymbol{\sigma}_i^2\}_{i=1}^M$ during training.

**(3) Iterative Refinement Module.** After randomly sampling initial slot conditions from their respective marginal distributions in the grounded slot dictionary $\mathbf{s}^0 \sim \mathfrak{S}(\mathbf{z}) \in \mathbb{R}^{K \times d_z}$, we iteratively refine the slot representations using slot attention as described in Section 3 and Algorithm 1. The subset of slot priors we sample from for each input corresponds to the $K$ respective codebook vectors which are closest to the output of the abstraction function $\mathbf{z}' = \mathcal{A}(\Phi_e(\mathbf{x}))$, as outlined in Equation 2.

*Remark (Slot Posterior):* The posterior distribution of the slots $\mathbf{s}^{t=T}$ at iteration $T$ given $\mathbf{x}$ is:

$$p(\mathbf{s}^T \mid \mathbf{x}) = \delta \left( \mathbf{s}^T - \prod_{t=1}^T \mathcal{H}_\theta \left( \mathbf{s}^{t-1}, f(g(\hat{\mathbf{q}}^{t-1}, \mathbf{k}), \mathbf{v})) \right) \right), \qquad (3)$$

where $\delta(\cdot)$ is Dirac delta distributed given randomly sampled initial slots from their marginals $\mathbf{s}^0 \sim p(\mathbf{s}^0 \mid \tilde{\mathbf{z}}) = \prod_{i=1}^K \mathcal{N}\left( \mathbf{s}_i^0; \boldsymbol{\mu}_i, \boldsymbol{\sigma}_i^2 \right)$, associated with the codebook vectors $\tilde{\mathbf{z}} \sim q(\tilde{\mathbf{z}} \mid \mathbf{x})$. The distribution over the initial slots $\mathbf{s}^0$ induces a distribution over the refined slots $\mathbf{s}^T$. One important aspect worth re-stating here is that the sampling of the initial slots $\left\{ \mathbf{s}_i^0 \sim p(\mathbf{s}_i^0 \mid \tilde{\mathbf{z}}_i) \right\}_{i=1}^K \subset \mathfrak{S}^2$ depends on the *indices* of the associated codebook vectors $\tilde{\mathbf{z}} \subset \mathfrak{S}^1$, as outlined in Definition 1, and is not conditioned on the values of $\tilde{\mathbf{z}}$ themselves. In proposition 3, in App. we demonstrate the convergence of slot prior distributions to the mean of true slot distributions, using the structural identifiability of the model under the assumptions 1-5.

---

**Algorithm 1** Conditional Slot Attention (CoSA).

1: **Input:** inputs $\mathbf{z} = \Phi_e(\mathbf{x}) \in \mathbb{R}^{N \times d_z}$, $\mathbf{k} = \mathcal{K}_\beta(\mathbf{z}) \in \mathbb{R}^{N \times d_s}$, and $\mathbf{v} = \mathcal{V}_\phi(\mathbf{z}) \in \mathbb{R}^{N \times d_s}$
2: **Spectral Decomposition:**
3:     $\mathbf{z} \cdot \mathbf{z}^T = \boldsymbol{V} \Lambda \boldsymbol{V}^T$             $\triangleright \boldsymbol{V}$: eigenvectors, $\Lambda = \mathrm{diag}(\lambda_i)$: eigenvalues
4:     $\mathbf{z}' = \boldsymbol{V}^T \cdot \mathbf{z}$                    $\triangleright \mathbf{z}'$: project onto principle components
5:     $K = 1 + \sum_{i=2}^{\tilde{N}} \lceil \lfloor \lambda_i \rfloor / \lambda_s \rceil$         $\triangleright$ Dynamically estimate number of slots
6: **for** $i = 0, \ldots, R$                         $\triangleright R$ Monte Carlo samples
7:     $\mathrm{slots}_i^0 \sim \mathfrak{S}(\mathbf{z}') \in \mathbb{R}^{K \times d_s}$           $\triangleright$ Sample $K$ initial slots from GSD
8:     **for** $t = 1, \ldots, T$               $\triangleright$ Refine slots over $T$ attention iterations
9:        $\mathrm{slots}_i^t = f\left(g\left(\mathcal{Q}_\gamma\left(\mathrm{LayerNorm}\left(\mathrm{slots}_i^{t-1}\right)\right), \mathbf{k}\right), \mathbf{v}\right)$    $\triangleright$ Update slot representations
10:       $\mathrm{slots}_i^t \mathrel{+}= \mathrm{MLP}\left(\mathrm{LayerNorm}\left(\mathcal{H}_\theta\left(\mathrm{slots}_i^{t-1}, \mathrm{slots}_i^t\right)\right)\right)$    $\triangleright$ GRU update & skip connection
11: **return** $\sum_i \mathrm{slots}_i / R$                    $\triangleright$ MC estimate

---

④ **Discovery Module.** For object discovery and scene composition tasks, we need to translate object-level slot representations into the observational space of the data. To achieve this, we use a spatial broadcast decoder (**?**), as employed by IODINE and slot attention (Greff et al., 2019; Locatello et al., 2020). This decoder reconstructs the image $\mathbf{x}$ via a softmax combination of $K$ individual reconstructions from each slot. Each reconstructed image $\mathbf{x}_s = \Phi_d(\mathbf{s}_k)$ consists of four channels: RGB plus a mask. The masks are normalized over the number of slots $K$ using softmax, therefore they represent whether each slot is bound to each pixel in the input.

We can now define a generative model of $\mathbf{x}$ which factorises as: $p(\mathbf{x}, \mathbf{s}^0, \tilde{\mathbf{z}}) = p(\mathbf{x} \mid \mathbf{s}^0)p(\mathbf{s}^0 \mid \tilde{\mathbf{z}})p(\tilde{\mathbf{z}})$, recalling that $\mathbf{s}^0 := \mathbf{s}_1^0, \ldots, \mathbf{s}_K^0$ are the initial slots at iteration $t = 0$, and $\tilde{\mathbf{z}}$ are our discrete latent variables described in Definition 1. For training, we derive a variational lower bound on the marginal log-likelihood of $\mathbf{x}$, a.k.a. the Evidence Lower Bound (ELBO), as detailed in Proposition 1.

**Proposition 1** (ELBO for Object Discovery). *Under a categorical distribution over our discrete latent variables $\tilde{\mathbf{z}}$, and the object-level prior distributions $p(\mathbf{s}_i^0) = \mathcal{N}\left(\mathbf{s}_i^0; \boldsymbol{\mu}_i, \boldsymbol{\sigma}_i^2\right)$ contained in $\mathfrak{S}^2$, we show that variational lower bound on the marginal log-likelihood of $\mathbf{x}$ can be expressed as:*

$$\log p(\mathbf{x}) \geq \mathbb{E}_{\tilde{\mathbf{z}} \sim q(\tilde{\mathbf{z}} \mid \mathbf{x}), \mathbf{s}^0 \sim p(\mathbf{s}^0 \mid \tilde{\mathbf{z}})} \left[\log p(\mathbf{x} \mid \mathbf{s})\right] - D_{\mathrm{KL}}\left(q\left(\tilde{\mathbf{z}} \mid \mathbf{x}\right) \| p(\tilde{\mathbf{z}})\right) =: \mathrm{ELBO}(\mathbf{x}), \quad (4)$$

*where $\mathbf{s} := \prod_{t=1}^T \mathcal{H}_\theta\left(\mathbf{s}^{t-1} \mid f(g(\hat{\mathbf{q}}^{t-1}, \mathbf{k}), \mathbf{v})\right)$ denotes the output of the iterative refinement procedure described in Algorithm 1 applied to the initial slot $\mathbf{s}^0$ representations.*

The proof is given in App. B.

⑤ **Reasoning Module.** The reasoning tasks we consider consist of classifying images into rule-based classes $\mathbf{y} \in \mathcal{Y}_1$, while providing a *rationale* $\mathbf{p}$ behind the predicted class. For example, class A might be something like "large cube and large cylinder", and a rationale for a prediction emerges directly from the slots binding to an input cube and cylinder. Refer to App. C for more details.

To achieve this, we define a set of learned functions $\mathfrak{S}^3 := \left\{\mathfrak{S}_1^3, \ldots, \mathfrak{S}_M^3\right\}$, which map refined slot representations $\mathbf{s}^{t=T}$ to $M$ *rationales* for each object *type* in the grounded slot dictionary $\mathfrak{S}$. This expands the dictionary to three elements: $\mathfrak{S} := \left\{(\mathfrak{S}^1, \mathfrak{S}^2, \mathfrak{S}^3)\right\}$. The reasoning task predictor $\Phi_r$ combines $K$ rationales extracted from $K$ slot representations of an input, which we denote by $\mathbf{p}_i = \mathfrak{S}_i^3(\mathbf{s}_i^T) \in \mathbb{R}^{|\mathcal{Y}_2|}$, and maps them to the rule-based class labels. The optimization objective for this reasoning task is a variational lower bound on the conditional log-likelihood, as in Proposition 2.

**Proposition 2** (ELBO for Reasoning Tasks). *Under a categorical distribution over our discrete latent variables $\tilde{\mathbf{z}}$, and the object-level prior distributions $p(\mathbf{s}_i^0) = \mathcal{N}\left(\mathbf{s}_i^0; \boldsymbol{\mu}_i, \boldsymbol{\sigma}_i^2\right)$ contained in $\mathfrak{S}^2$, the variational lower bound on the conditional log-likelihood of $\mathbf{y}$ given $\mathbf{x}$ is given by:*

$$\log p(\mathbf{y} \mid \mathbf{x}) \geq \mathbb{E}_{\tilde{\mathbf{z}} \sim q(\tilde{\mathbf{z}} \mid \mathbf{x}), \mathbf{s}^0 \sim p(\mathbf{s}^0 \mid \tilde{\mathbf{z}})} \left[\log p(\mathbf{y} \mid \mathbf{s}, \mathbf{x})\right] - D_{\mathrm{KL}}\left(q(\tilde{\mathbf{z}} \mid \mathbf{x}) \| p(\tilde{\mathbf{z}})\right). \quad (5)$$

*noting that conditioning $\mathbf{y}$ on $\{\mathbf{s}, \mathbf{x}\}$ is equivalent to conditioning on the predicted rationales $\mathbf{p}$, since the latter are deterministic functions of the former.*

The proof is similar to proposition 1 and is given in App. B for completeness.

Table 1: Foreground ARI on CLEVR, Tetrominoes, ObjectsRoom, and COCO datasets, for all baseline models and CoSA-cosine variant. For COCO we use the DINOSAUR variant (Seitzer et al., 2022) (with ViT-S16 as feature extractor) as a baseline model and use the same architecture for IMPLICT and CoSA (adaptation details in App. H.3).

| METHOD | TETROMINOES | CLEVR | OBJECTSROOM | COCO |
|---|---|---|---|---|
| SA (Locatello et al., 2020) | $0.99 \pm 0.005$ | $0.93 \pm 0.002$ | $0.78 \pm 0.02$ | - |
| BlockSlot (Singh et al., 2022) | $0.99 \pm 0.001$ | $0.94 \pm 0.001$ | $0.77 \pm 0.01$ | - |
| SlotVAE (Wang et al., 2023) | - | - | $0.79 \pm 0.01$ | - |
| DINOSAUR (Seitzer et al., 2022) | - | - | - | $0.28 \pm 0.02$ |
| IMPLICIT (Chang et al., 2022) | $0.99 \pm 0.001$ | $0.93 \pm 0.001$ | $0.78 \pm 0.003$ | $0.28 \pm 0.01$ |
| CoSA | $0.99 \pm 0.001$ | $\mathbf{0.96} \pm 0.002$ | $\mathbf{0.83} \pm 0.002$ | $\mathbf{0.32} \pm 0.01$ |

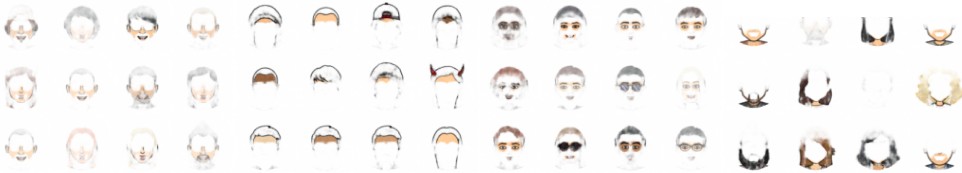

(a) Embedding-7 ($\mathfrak{S}^1_7$)   (b) Embedding-14 ($\mathfrak{S}^1_{14}$)   (c) Embedding-25 ($\mathfrak{S}^1_{25}$)   (d) Embedding-55 ($\mathfrak{S}^1_{55}$)

Figure 3: GSD binding: we can observe that *cheeks* being bound to $\mathfrak{S}^1_7$, *forehead* to $\mathfrak{S}^1_{14}$, *eyes* to ($\mathfrak{S}^1_{25}$), and *facial hair* to $\mathfrak{S}^1_{55}$, illustrating the notion of object binding achieved in GSD, in the case of bitmoji dataset for CoSA model trained with cosine sampling stratergy.

## 5 EXPERIMENTS

Our empirical study evaluates CoSA on a variety of popular *object discovery* (Table 1, 2) and *visual reasoning* benchmarks (Table 3). In an *object discovery* context, we demonstrate our method's ability to: (i) dynamically estimate the number of slots required for each input (Fig. 4); (ii) map the grounded slot dictionary elements to particular object *types* (Fig. 3); (iii) perform reasonable slot composition without being explicitly trained to do so (Fig. 5). In terms of *visual reasoning*, we show the benefits of grounded slot representations for generalization across different domains, and for improving the quality of generated rationales behind rule-based class prediction (Table 3). We also provide detailed ablation studies on: (i) the on choice of codebook sampling (App. H); (ii) different codebook regularisation techniques to avoid collapse (App. G); (iii) the choice of abstraction function (App. I). For details on the various datasets we used, please refer to App. C. For training details, hyperparameters, initializations and other computational aspects refer App. M.

### 5.1 CASE STUDY 1: OBJECT DISCOVERY & COMPOSITION

In this study, we provide a thorough evaluation of our framework for object discovery across multiple datasets, including: CLEVR (Johnson et al., 2017), Tetrominoes (Kabra et al., 2019), Objects-Room (Kabra et al., 2019), Bitmoji (Mozafari, 2020), FFHQ (Karras et al., 2020), and COCO (Lin et al., 2014). In terms of evaluation metrics, we use the foreground-adjusted Rand Index score (ARI) and compare our method's results with standard SA (Locatello et al., 2020), SlotVAE (Wang et al., 2023), BlockSlot (Singh et al., 2022), IMPLICIT (Chang et al., 2022) and DINO (Seitzer et al., 2022). We evaluated the majority of the baseline models on the considered datasets using their original implementations, with the exception of SlotVAE, where we relied on the original results from the paper. For DINOSAUR, we specifically assessed its performance on a real-world COCO dataset, following the code adaptation outlined in App. H.3. Additionally, we measured the reconstruction score using MSE and the FID (Heusel et al., 2017). We analyse three variants of CoSA: Gumbel, Euclidian, and Cosine, corresponding to three different sampling strategies used for the GSD. Detailed analysis and ablations on different types of abstraction function are detailed in App. I.

The inclusion of GSD provides two notable benefits: (i) it eliminates the need for a hard requirement of parameter $K$ during inference, and (ii) since slots are initialized with samples from their grounded counterparts, this leads to improved initialization and allows us to average across multiple samples, given that the slots are inherently aligned. These design advantages contribute to the enhanced

Table 2: Object discovery results on multiple benchmarking datasets and previously existing methods. We measure both MSE and FID metrics to compare the results.

| METHOD | CLEVR | | TETROMINOES | | OBJECTS-ROOM | | BITMOJI | | FFHQ | |
|---|---|---|---|---|---|---|---|---|---|---|
| | MSE ↓ | FID ↓ | MSE ↓ | FID ↓ | MSE ↓ | FID ↓ | MSE ↓ | FID ↓ | MSE ↓ | FID ↓ |
| SA | 6.37 | 38.18 | 1.49 | 3.81 | 7.57 | 38.12 | 14.62 | 12.78 | 55.14 | 54.95 |
| Block-Slot | 3.73 | 34.11 | 0.48 | 0.42 | 5.28 | 36.33 | 10.24 | 11.01 | 52.16 | 41.56 |
| IMPLICIT | 3.95 | 38.16 | 0.59 | 0.41 | 5.56 | 36.48 | 9.87 | 10.68 | 47.06 | 49.95 |
| CoSA | **3.14** | **29.12** | **0.42** | 0.41 | **4.85** | **28.19** | **8.17** | **9.28** | **33.37** | **36.34** |

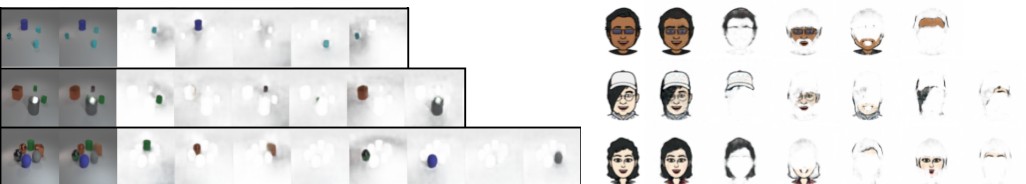

Figure 4: Object discovery: reconstruction quality and dynamic slot number selection for CoSA-COSINE on CLEVR and Bitmoji, with an MAE of **2.06** over slot number estimation for CLEVR.

representational quality for object discovery, as evidenced in Table 1 and 2, while also enhancing generalizability which will be demonstrated later. To evaluate the quality of the generated slots without labels, we define a some additional metrics like the overlapping index (OPI) and average slot FID (SFID) (ref. Section D, Table 9).

Fig. 4 showcases the reconstruction quality of CoSA and its adaptability to varying slot numbers. For additional results on all datasets and a method-wise qualitative comparison, see App. H. Regarding the quantitative assessment of slot number estimation, we calculate the mean absolute error (MAE) by comparing the estimated slot count to the actual number of slots. Specifically, the MAE values for the CLEVR, Tetrominoes, and Objects-Room datasets are observed to be **2.06, 0.02**, and **0.32**, respectively. On average, dynamic slot number estimation reduces **38%, 26%**, and **23%** FLOPs on CLEVR, Objects-Room, and Tetrominoes as opposed to fixed number of slots. In Fig. 3, we illustrate the grounded object representations within GSD, which are obtained by visualising the obtained slots as a result of sampling a particular element in the GSD. Furthermore, we explore the possibility of generating novel scenes using the learned GSD, although out models were not designed explicitly for this purpose. To that end, we sample slots from randomly selected slot distributions and decode them to generate new scenes. The prompting in our approach relies solely on the learned GSD, and we do not need to construct a prompt dictionary like SLATE (Singh et al., 2021). To evaluate scene generation capabilities, we compute FID and SFID scores on randomly generated scenes across all datasets; please refer to App. Table 14. Fig. 5 provides visual representations of images generated through slot composition on CLEVR, Bitmoji, and Tetrominoes datasets.

## 5.2 CASE STUDY 2: VISUAL REASONING & GENERALIZABILITY

As briefly outlined in the reasoning module description (Section 4), the visual reasoning task involves classifying images into rule-based classes, while providing a rationale behind the predicted class. To evaluate and compare our models, we use the F1-score, accuracy, and the Hungarian Matching Coefficient (HMC), applied to benchmark datasets: CLEVR-Hans3, CLEVR-Hans7 (Stammer et al., 2021), FloatingMNIST-2, and FloatingMNIST-3. CLEVR-Hans includes both a confounded validation set and a non-confounded test set, enabling us to verify the generalizability of our model. FloatingMNIST (FMNIST) is a variant of the MNIST (Deng, 2012) dataset with three distinct reasoning tasks, showcasing the model's adaptability across domains. Further details on the datasets are given in App. C.

To facilitate comparisons, we train and evaluate several classifiers, including: (i) the baseline classifier; (ii) a classifier with a vanilla SA bottleneck; (iii) Block-Slot; and (iv) our CoSA classifiers. The main comparative results are presented in Table 3, and additional results for FloatingMNIST3 and CLEVR-Hans datasets are available in the App. (Tables 16-19). We conduct task adaptability

Table 3: Accuracy and Hungarian matching coefficient (HMC) for reasoning tasks on addition and subtraction variant of FMNIST2 dataset. Here, the first and third pair of columns correspond to the models trained and tested on the FMNIST2-Add and FMNIST2-Sub datasets, respectively, while the second and fourth pair correspond to few-shot ($k$=100) adaptability results across datasets.

| METHOD | FMNIST2-ADD$_{source}$ | | FMNIST2-SUB$_{target}$ | | FMNIST2-SUB$_{source}$ | | FMNIST2-ADD$_{target}$ | |
|---|---|---|---|---|---|---|---|---|
| | ACC ↑ | HMC ↓ | ACC ↑ | F1 ↑ | ACC ↑ | HMC ↓ | ACC ↑ | F1 ↑ |
| CNN | 97.62 | - | 10.35 | 10.05 | 98.16 | - | 12.35 | 9.50 |
| SA | 97.33 | 0.14 | 11.06 | 09.40 | 97.41 | 0.13 | 08.28 | 7.83 |
| Block-Slot | 98.11 | 0.12 | 09.71 | 09.10 | 97.42 | 0.14 | 09.61 | 8.36 |
| CoSA | **98.12** | **0.10** | **60.24** | **50.16** | **98.64** | **0.12** | **63.29** | **58.29** |

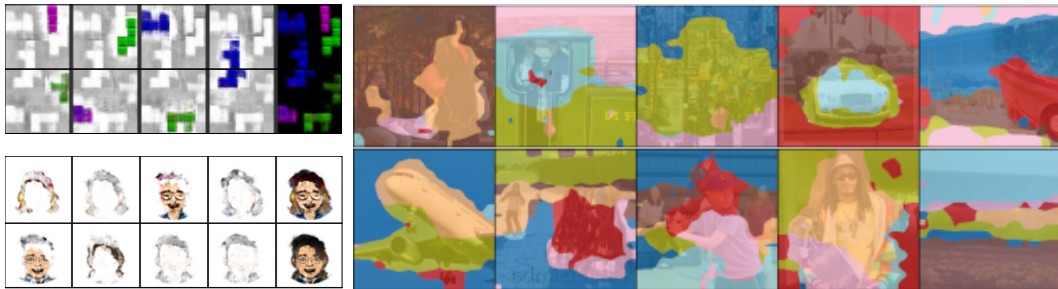

Figure 5: Top and bottom left illustrates the randomly prompted slots and their composition. Right demonstrates object discovery results of CoSA on COCO dataset.

experiments to assess the reusability of grounded representations in our learned GSD. To that end, we create multiple variants of the FloatingMNIST dataset, introducing addition, subtraction, and mixed tasks. Initially, we train the model with one specific objective, which we consider as the source dataset and assess its capacity to adapt to other target datasets by fine-tuning the task predictor layer through $k$-shot training. The adaptability results for FloatingMNIST-2 are provided in Table 3, and results for FloatingMNIST-3 are detailed in the App. in Table 19. Results for mixed objectives are also discussed in App. L. Fig. 28-30 depict the few-shot adaptation capabilities over multiple $k$ iterations on the FloatingMNIST2, FloatingMNIST3, and CLEVR-Hans datasets.

Overall we observe that GSD helps in (i) better capturing the rational for the predicted class, as illustrated by HMC measure; and (ii) learning reusable properties leading to better generalisability as demonstrated by accuracy and F1 measure in $k$-shot adaptability tasks; detailed in Table 3, 16- 18.

## 6 CONCLUSION

In this work, we propose conditional slot attention (CoSA) using a grounded slot dictionary (GSD). CoSA allows us to bind arbitrary instances of an object to a specialized canonical representation, encouraging invariance to identity-preserving changes. This is in contract to vanilla slot attention which presupposes a *single* distribution from which all slots are randomly initialised. Additionally, our proposed approach enables dynamic estimation of the number of required slots for a given scene, saving up to **38%** of FLOPs. We show that the grounded representations allow us to perform random slot composition without the need of constructing a prompt dictionary as in previous works. We demonstrate the benefits of grounded slot representations in multiple downstream tasks such as scene generation, composition, and task adaptation, whilst remaining competitive with SA in popular object discovery benchmarks. Finally, we demonstrate adaptability of grounded representations resulting up to **5x** relative improvement in accuracy compared to SA and standard CNN architectures.

The main limitations of the proposed framework include: (i) limited variation in slot-prompting and composition; (ii) assumption of no representation overcrowding; (iii) maximum object area assumption. (App. N). An interesting future direction would be to use a contrastive learning approach to learn a dictionary for disentangled representation of position, shape, and texture, resulting in finer control over scene composition. From a reasoning point of view, it would be interesting to incorporate background knowledge in the framework, which could aid learning exceptions to rules.

## 7 BROADER IMPACT

Our proposed method allows us to learn conditional object-centric distributions, which have wide applications based on the selected downstream tasks. Here, we demonstrate its working on multiple controlled environments. The scaled versions with a lot more data and compute can generalize across domains resulting in foundational models for object-centric representation learning; the negative societal impacts (like: realistic scene composition/object swapping from a given scene) for such systems should carefully be considered, while more work is required to properly address these concerns. As demonstrated in the reasoning task, our method adapts easily to different downstream tasks; if we can guarantee the human understandability of learned properties in the future, our method might be a step towards learning interpretable and aligned AI models.

## 8 ACKNOWLEDGEMENTS

A. Kori is supported by UKRI (grant agreement no. EP/S023356/1), as part of the UKRI Centre for Doctoral Training in Safe and Trusted AI. F.D.S. Ribeiro and B. Glocker received funding from the European Research Council (ERC) under the European Union's Horizon 2020 research and innovation programme (grant agreement No 757173, project MIRA, ERC-2017-STG).

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

# Appendix

## Table of Contents

# A ASSUMPTIONS

Now, we list all the assumptions for modeling our slot dictionary and also discuss the how some of these assumptions are implicitly made in previous baselines (ref. Table 4.

**Assumption 1** (Representation separation). Object level separation present in observational space can be observed in the latent space of the model (*i.e.,* if $\mathbf{x} = \bigcup_i^K \{O_i\}$, then $\mathbf{z} = \bigcup_i^K \{Oz_i\}$, where $O_i$ and $Oz_i$ correspond to individual object representation in observational and latent space respectively).

*Remark* 1. In practice, when features are learned with an end-to-end objective, object-level representation emerges due to iterative attention.

**Assumption 2** (Representation overcrowding). The latent representation in $\mathbf{z} \in \mathbb{R}^{N \times d_z}$ can accurately recover $K$ slots when $K \leq N$. In other words, $\mathbf{z}_i \in \mathbb{R}^{d_z}$ is an element of $\mathbf{z}$ that can only encode the knowledge of a single object.

*Remark* 2. Verifying this property is difficult. However, the latent layer for CoSA can be empirically estimated to minimize this effect.

**Assumption 3** (Object level sufficiency). We assume that there are no additional objects in the original data distributions other than the ones expressed in training data.

*Remark* 3. The assumption on object level sufficiency may not be required in SA. Here, this is required as the aim is to learn marginal distributions for every known object in the dataset.

**Assumption 4** (Sufficient dictionary components). We assume the slot dictionary to have a sufficient number $\tilde{M}$ of $(\mathfrak{S}^1, \mathfrak{S}^2)$ pairs to capture the emergent behavior of objectness in the entire dataset, such that $\tilde{M} \geq K + 1$.

**Assumption 5** (Injectivity Assumption). We assume the resulting decoder model is injective as by construction the architecture follows the properties as expressed in Khemakhem et al. (2020).

*Remark* 4. The assumption is mainly used for the theoretical result in proposition 3, in we can observe non distinct object representations in the generated scene if this property is not satisfied.

Table 4: Comparison between methods and their implicit assumptions and tasks achieved.

| METHOD | DATA | ASSUMPTION | TASKS | GROUNDING |
|---|---|---|---|---|
| Engelcke et al. (2021; 2019); Burgess et al. (2019); Wang et al. (2023); Emami et al. (2022) | Image Data | A1, A2 | Representation, Segregation, and Generation | No |
| Locatello et al. (2020); Chang et al. (2022); Seitzer et al. (2022); | Image Data | A1, A2 | Representation, Segregation | No |
| Elsayed et al. (2022); Kipf et al. (2021); | Image Data and conditioning info. | A1, A2, A3 | Representation, Segregation | No |
| Van Steenkiste et al. (2020); Lin et al. (2020); Singh et al. (2021) | Image Data | A1, A2, A3 | Representation, Segregation, and Generation (constr--ained environment or predefined prompts) | No |
| CoSA | Image Data | A1, A2, A3, A4 | Representation, Segregation, and Generation | Yes |

# B PROOFS

## B.1 PROPOSITION 1: *(Object Discovery - ELBO formulation)*:

Under a categorical distribution over our discrete latent variables $\tilde{\mathbf{z}}$, and the object-level prior distributions $p(\mathbf{s}_i^0) = \mathcal{N}\left(\mathbf{s}_i^0; \boldsymbol{\mu}_i, \boldsymbol{\sigma}_i^2\right)$ contained in $\mathfrak{S}^2$, we show that variational lower bound on the

marginal log-likelihood of $\mathbf{x}$ can be expressed as:

$$\log p(\mathbf{x}) \geq \mathbb{E}_{\tilde{\mathbf{z}} \sim q(\tilde{\mathbf{z}} | \mathbf{x}), \mathbf{s}^0 \sim p(\mathbf{s}^0 | \tilde{\mathbf{z}})} \left[ \log p(\mathbf{x} \mid \mathbf{s}) \right] - D_{\mathrm{KL}} \left( q \left( \tilde{\mathbf{z}} \mid \mathbf{x} \right) \right) \| p(\tilde{\mathbf{z}})) =: \mathrm{ELBO}(\mathbf{x}), \quad (6)$$

where $\mathbf{s} := \prod_{t=1}^{T} \mathcal{H}_\theta \left( \mathbf{s}^{t-1} \mid f(g(\hat{\mathbf{q}}^{t-1}, \mathbf{k}), \mathbf{v}) \right)$ denotes the output of the iterative refinement procedure described in Algorithm 1 applied to the initial slot $\mathbf{s}^0$ representations.

*Proof.* For this proof, we consider the data distribution as $p(\mathbf{x})$, and the aim is to maximize the log-likelihood of this distribution:

$$\log p(\mathbf{x}) = \log \int_{\mathbf{s}} \int_{\tilde{\mathbf{z}}} p(\mathbf{x}, \mathbf{s}, \tilde{\mathbf{z}}) d\mathbf{s} d\tilde{\mathbf{z}}$$

Consider variational distributions $q(\tilde{\mathbf{z}} \mid \mathbf{x})$.

$$= \log \int_{\mathbf{s}} \int_{\tilde{\mathbf{z}}} p(\mathbf{x}, \mathbf{s}, \tilde{\mathbf{z}}) \frac{q(\tilde{\mathbf{z}} \mid \mathbf{x})}{q(\tilde{\mathbf{z}} \mid \mathbf{x})} d\mathbf{s} d\tilde{\mathbf{z}}$$

$$\geq \mathbb{E}_{\tilde{\mathbf{z}} \sim q(\tilde{\mathbf{z}} | \mathbf{x})} \mathbb{E}_{\mathbf{s}^0 \sim p(\mathbf{s}^0 | \tilde{\mathbf{z}})} \log \frac{p(\mathbf{x} \mid \mathbf{s}) p(\tilde{\mathbf{z}})}{q(\tilde{\mathbf{z}} \mid \mathbf{x})}$$

$$= \mathbb{E}_{\tilde{\mathbf{z}} \sim q(\tilde{\mathbf{z}} | \mathbf{x})} \mathbb{E}_{\mathbf{s}^0 \sim p(\mathbf{s}^0 | \tilde{\mathbf{z}})} \log \frac{p(\tilde{\mathbf{z}})}{q(\tilde{\mathbf{z}} \mid \mathbf{x})} + \mathbb{E}_{\tilde{\mathbf{z}} \sim q(\tilde{\mathbf{z}} | \mathbf{x})} \mathbb{E}_{\mathbf{s}^0 \sim p(\mathbf{s}^0 | \tilde{\mathbf{z}})} \log p(\mathbf{x} \mid \mathbf{s})$$

$$= \mathbb{E}_{\tilde{\mathbf{z}} \sim q(\tilde{\mathbf{z}} | \mathbf{x})} \log \frac{p(\tilde{\mathbf{z}})}{q(\tilde{\mathbf{z}} \mid \mathbf{x})} + \mathbb{E}_{\tilde{\mathbf{z}} \sim q(\tilde{\mathbf{z}} | \mathbf{x})} \mathbb{E}_{\mathbf{s}^0 \sim p(\mathbf{s}^0 | \tilde{\mathbf{z}})} \log p(\mathbf{x} \mid \mathbf{s})$$

$$= \mathbb{E}_{\tilde{\mathbf{z}} \sim q(\tilde{\mathbf{z}} | \mathbf{x})} \mathbb{E}_{\mathbf{s}^0 \sim p(\mathbf{s}^0 | \tilde{\mathbf{z}})} \log p(\mathbf{x} \mid \mathbf{s} = g(\mathbf{s}^0)) - D_{\mathrm{KL}} \Big( q(\tilde{\mathbf{z}} \mid \mathbf{a} = (\mathcal{A} \circ \Phi_e)(\mathbf{x})) \| p(\tilde{\mathbf{z}}) \Big)$$

$$\square$$

## B.2 PROPOSITION 2: *(ELBO formulation for reasoning task)*:

Under a categorical distribution over our discrete latent variables $\tilde{\mathbf{z}}$, and the object-level prior distributions $p(\mathbf{s}_i^0) = \mathcal{N} \left( \mathbf{s}_i^0; \boldsymbol{\mu}_i, \boldsymbol{\sigma}_i^2 \right)$ contained in $\mathbb{S}^2$, the variational lower bound on the conditional log-likelihood of $\mathbf{y}$ given $\mathbf{x}$ is given by:

$$\log p(\mathbf{x}) \geq \mathbb{E}_{\tilde{\mathbf{z}} \sim q(\tilde{\mathbf{z}} | \mathbf{x})} \mathbb{E}_{\mathbf{s}^0 \sim p(\mathbf{s}^0 | \tilde{\mathbf{z}})} \log p(\mathbf{y} \mid \mathbf{p}) - D_{\mathrm{KL}} \Big( q(\tilde{\mathbf{z}} \mid \mathbf{x}) \| p(\tilde{\mathbf{z}}) \Big). \quad (7)$$

*Proof.* The proof is very similar to the proof of proposition 1 . We include this for the sake of completion. For this, we consider categorical conditional distribution as $p(\mathbf{y} \mid \mathbf{x})$, the model segregating slots into given categories. The aim is to maximize the log-likelihood of this distribution:

$$\log p(\mathbf{y} \mid \mathbf{x}) = \log \int_{\mathbf{s}} \int_{\tilde{\mathbf{z}}} p(\mathbf{y}, \mathbf{s}, \tilde{\mathbf{z}} \mid \mathbf{x}) d\mathbf{s} d\tilde{\mathbf{z}}$$

Consider variational distributions $q(\tilde{\mathbf{z}} \mid \mathbf{x})$.

$$= \log \int_{\mathbf{s}} \int_{\tilde{\mathbf{z}}} p(\mathbf{y}, \mathbf{s}, \tilde{\mathbf{z}} \mid \mathbf{x}) \frac{q(\tilde{\mathbf{z}} \mid \mathbf{x})}{q(\tilde{\mathbf{z}} \mid \mathbf{x})} d\mathbf{s} d\tilde{\mathbf{z}}$$

$$\geq \mathbb{E}_{\tilde{\mathbf{z}} \sim q(\tilde{\mathbf{z}} | \mathbf{x})} \mathbb{E}_{\mathbf{s}^0 \sim p(\mathbf{s}^0 | \tilde{\mathbf{z}})} \log \frac{p(\mathbf{y} \mid \mathbf{x}, \mathbf{s}^0) p(\tilde{\mathbf{z}})}{q(\tilde{\mathbf{z}} \mid \mathbf{x})}$$

$$= \mathbb{E}_{\tilde{\mathbf{z}} \sim q(\tilde{\mathbf{z}} | \mathbf{x})} \mathbb{E}_{\mathbf{s}^0 \sim p(\mathbf{s}^0 | \tilde{\mathbf{z}})} \left[ \log \frac{p(\tilde{\mathbf{z}})}{q(\tilde{\mathbf{z}} \mid \mathbf{x})} \right] + \mathbb{E}_{\tilde{\mathbf{z}} \sim q(\tilde{\mathbf{z}} | \mathbf{x})} \mathbb{E}_{\mathbf{s}^0 \sim p(\mathbf{s}^0 | \tilde{\mathbf{z}})} \log p(\mathbf{y} \mid \mathbf{p})$$

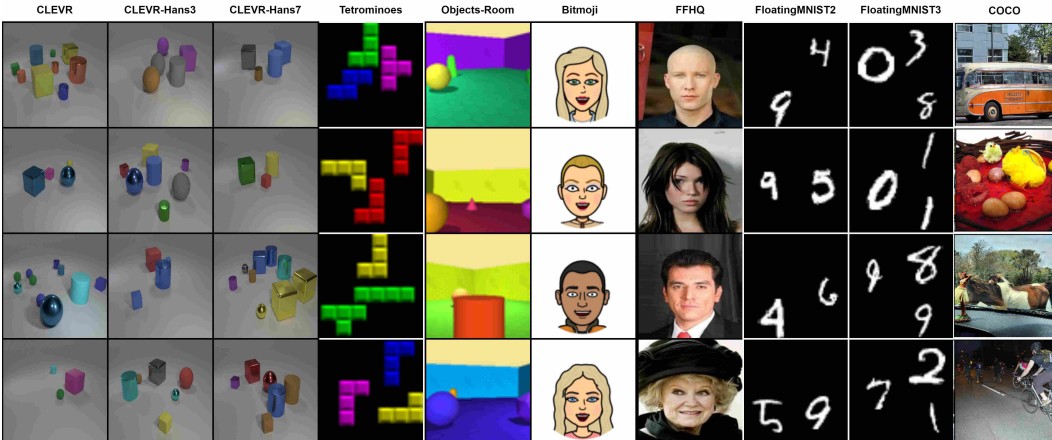

Figure 6: This image overviews all the datasets used in this work, from left to right columns correspond to CLEVR, CLEVR-Hans3, CLEVR-Hans7, Tetrominoes, Objects-Room, Bitmoji, FFHQ, FloatingMNIST-2, FloatingMNIST-3, and COCO datasets respectively.

$$= \mathbb{E}_{\tilde{\mathbf{z}} \sim q(\tilde{\mathbf{z}}|\mathbf{x})} \left[ \log \frac{p(\tilde{\mathbf{z}})}{q(\tilde{\mathbf{z}} \mid \mathbf{x})} \right] + \mathbb{E}_{\tilde{\mathbf{z}} \sim q(\tilde{\mathbf{z}}|\mathbf{x})} \mathbb{E}_{\mathbf{s}^0 \sim p(\mathbf{s}^0|\tilde{\mathbf{z}})} \log p(\mathbf{y} \mid \mathbf{p})$$

$$= \mathbb{E}_{\tilde{\mathbf{z}} \sim q(\tilde{\mathbf{z}}|\mathbf{x})} \mathbb{E}_{\mathbf{s}^0 \sim p(\mathbf{s}^0|\tilde{\mathbf{z}})} \log p(\mathbf{y} \mid \mathbf{p}) - D_{\mathrm{KL}}\Big( q(\tilde{\mathbf{z}} \mid \mathbf{a} = (\mathcal{A} \circ \Phi_e)(\mathbf{x})) \mid\mid p(\tilde{\mathbf{z}}) \Big)$$

$\square$

## C  DATASETS

In this work, we use multiple datasets for every case studies. We make use of publically available datasets that are released under MIT Licence and that are open for all research works. We also created two new variants of the MNIST dataset called FloatingMNIST2 (FMNIST2) and FloatingMNIST3 (FMNIST3) for evaluation reasoning/discriminative tasks with the proposed method. We will release the data-generation scripts along with the source code of this project under the open-source MIT License. Table 5 describes maximum number of objects per image in the considered object discovery datasets. We details on the dataset specifications in this section:

### C.1  CLEVR (JOHNSON ET AL., 2017)

CLEVR is diagnostic dataset usually used for benchmarking visual question answering tasks. Here, in our tasks we don't use the question answering part of the dataset, we make use this dataset for its object level compsitionality aspect. This dataset consists of 70000, 15000, and 15000 set of training, validation, and testing images respectively. In our tasks we resize all the images to 64x64 dimension, normalizing them to bound pixel intensities to lie between [0, 1].

### C.2  CLEVR-HANS3 (STAMMER ET AL., 2021)

CLEVR-Hans is a dataset with multiple confouding factors, developed with the soul purpose of investigating reasoning possibilities within the network. The dataset was built on top of CLEVR dataset, where the CLEVR images were further categorized into multiple classes based on object attributes. Images in a particular class are encoded with confounded information with respect some attributes. CLEVR-Hans has in total of 3 classes with the following rules for class1: *Large cube and Large cylinder*, class2: *Small metal cube and Small sphere*, and class3: *Large blue sphere and small yellow sphere*. This dataset contains total of 9000, 2250, and 2250 images in training, validation, and testing set respectively (equally split between different classes). In our tasks we resize all the images to 64x64 dimension, normalizing them to bound pixel intensities to lie between [0, 1].

### C.3  CLEVR-HANS7 (STAMMER ET AL., 2021)

Similar to CLEVR-Hans3 CLEVR-Hans7 is built on top of CLEVR dataset, where the images are categorized into 7 classes based on the object attributes. Here, are the rules used for classifying images: class1: *Large cube and Large cylinder*, class2: *Small metal cube and Small sphere*, class3: *cyan object in front of 2 red objects*, class4: *image with small green, brown, and purple objects with two other small objects*, class5: *3 spheres or 3 spheres with 3 metal cylinders*, class6: *3 metal cylinders*, and class7: *large blue sphere with small yellow sphere*. This dataset contains total of 21000, 5250, and 5250 images in training, validation, and testing set respectively (equally split between different classes). In our tasks we resize all the images to 64x64 dimension, normalizing them to bound pixel intensities to lie between [0, 1].

### C.4  TETROMINOES (KABRA ET AL., 2019)

Tetrominoes is a Tetris-like shapes dataset. This provides a test bed for non-overlapping self-supervised object discovery. Here, we use around 100000, 10000, and 10000 images for training, validation, and testing. Each image is resized to 32x32 dimension and normalized to bound the intensity values between [0,1]. Each image comprises 3 tetris shapes sampled from 17 unique shapes and orientations, where each object has one of red, green, blue, yellow, magenta, or cyan color.

### C.5  OBJECTS-ROOM (KABRA ET AL., 2019)

Objects room is a direct extention of 3Dshapes dataset Kim & Mnih (2018), built using MuJoCo environment Eslami et al. (2018). While the actual dataset has 3 different variants, we make use of single variant with identical object and room variant. In this setting 4-6 objects are placed in the room and have an identical, where the colors are randomly sampled. Here, in our work we use around 100000, 10000, and 10000 images for training, validation, and testing. Each image is resized to 64x64 dimension and normalized to bound the intensity values between [0,1].

### C.6  BITMOJI (MOZAFARI, 2020)

Bitmoji is cartoon faces dataset for bitmoji mobile application. The images in bitmoji dataset consists 5-point facial landmarks for both male and female faces. In our task we resize all the images to 64x64 dimension and normalize them to bound pixel intensities to lie between [0, 1]. This dataset consists of 4084 images, divided into sets of 3500, 292, and 292 images for training, validation, and testing respectively.

### C.7  FFHQ (KARRAS ET AL., 2020)

This dataset includes high quality human faces along with the attributes corresponding to facial features. FFHQ is real world extention of bitmoji datasets. This dataset consists of approximately 200k images of 128x128 resolution with 40 different binary attributes, and the task is to categorize images based on gender (0 = male; 1=female).

### C.8  COCO (LIN ET AL., 2014)

COCO is a large-scale object recognition dataset with scene-level caption information. It consists of everyday scenes with common objects. The dataset consists of 328k images with varied resolutions. In this work, we resize the images to $224 \times 224$ resolution and perform our analysis for scene decomposition. The dataset consists of 91 different objects with around 2.5 million masks.

### C.9  FLOATINGMNIST

FloatingMNIST is inspired by the Moving MNIST dataset (Srivastava et al., 2015) and is used to benchmark reasoning tasks in the proposed model. FloatingMNIST is made by randomly combining MNIST digits Deng (2012). The dataset has three main objectives: addition, subtraction, and mixed. The addition task is to estimate the sum of all the digits present in the given image, the subtraction task is to estimate the absolute difference between the digits (after ordering them in decreasing order),

and finally, the mixed task is to perform addition if the digits are greater than five else performing subtraction.

**FloatingMNIST-2:** In the case of FloatingMNIST-2, we combine 2 MNIST digits in a large canvas creating the image of 64x64. We propose three variants FloatingMNIST-2-Add, FloatingMNIST-2-Sub, and FloatingMNIST-2-Mixed datasets. The task in FloatingMNIST-2-Add is to estimate the add of 2 digits present in the image. In the case of FloatingMNIST-2-Sub, the task is to estimate difference between 2 digits present in the image. While, In the case of FloatingMNIST-2-Mixed, the task is to sum the digits if both the digits are less than 5 else to estimate the absolute difference between them. We use 60000, 10000, and 10000 training, validation, and testing images respectively.

**FloatingMNIST-3:** Similar to FloatingMNIST-2, this dataset is made of images with 3 digits. Even this version of dataset consists of three variants of reasoning datasets, namely, FloatingMNIST-3-Add, FloatingMNIST-3-Sub, and FloatingMNIST-3-Mixed. Where the task in FloatingMNIST-3-Add and FloatingMNIST-3-Sub is to estimate sum and absolute difference beween all the digits. In the case of FloatingMNIST-3-Mixed, all digits less than 5 are added while the digits greater than 5 are substracted from the final target. We use 60000, 10000, and 10000 training, validation, and testing images respectively.

Table 5: Maximum number of objects (including background) observed in a single image per dataset.

| DATASETS($\downarrow$) | Number of Objects |
|---|---|
| CLEVR | 11 |
| TETROMINOES | 4 |
| OBJECTSROOM | 6 |

# D    METRICS

All the task specific metrics are described in this section

## D.1    MEAN SQUARED ERROR (MSE)

To compute the quality of the reconstructed image we compute pixel wise mean squared distance between the original image and the ground truth image. This metric doesn't provide any information about how good the image is decomposed into objects. As the task is unsupervised in nature, we measure MSE with few other properties to comment on the goodness of decomposition. MSE is formally described in equation 8, where $N$ and $x^i$ correspond to a total number of images and a particular image in the dataset.

$$\text{MSE} = \frac{1}{N} \sum_i^N ||x^i - \Phi_d(\text{CoSA}(\Phi_e(x^i)))||_2^2 \tag{8}$$

## D.2    OVERLAPPING INDEX (OPI)

OPI measures the mean overlap of a particular slot with respect to every other slot across the given dataset. For computing the overlap we first rescale the pixel intensities in the estimated mask to lie between $(0, 1)$ by applying *min-max* normalization and later thresholded at 0.5. The normalization and thresholding will result in binary decomposed images, which can be used to compute average overlap. OPI is formally described in equation 9, where $N, S$, and $x^i_{s_j}$ correspond to a total number of images, slots, and a particular slot estimated for considered image $x^i$ respectively.

$$\text{OPI} = \frac{1}{N} \sum_i^N \frac{1}{S} \sum_j^S \frac{1}{S-j} \sum_k^j \frac{2(x^i_{s_j} \cap x^i_{s_k})}{x^i_{s_j} \cup x^i_{s_k}} \tag{9}$$

### D.3    CODEBOOK DIVERGENCE (CBD)

We use this metric to measure the distance between the codebook embeddings, higher distance indicates the effective usage of codebook embeddings. To measure this distance we first project all the embedding vectors onto the unit hypersphere and compute the average cosine distance between them. In practice this is done by computing the inner product between the embedding vectors, which is formally described in equation 10, where $K$ corresponds a total number of codebook embeddings and $\cos^{-1}$ is applied to get the angular distance between the embeddings.

$$\text{CBD} = \arccos \frac{1}{K} \sum_i^K \frac{1}{K-i} \sum_j^i \langle e_i, e_j \rangle \qquad (10)$$

### D.4    CODEBOOK PERPLEXITY (CBP)

Perplexity measure is a property that indicates the sampling efficiency of the codebook. The perplexity is higher if all the codebook embeddings are uniformly sampled. Lower perplexity is an indicator of codebook collapse. Formally the perplexity is estimated as described in equation 11, where $N, K$, and $\text{qidx}_j^i$ correspond to a total number of images in a dataset, the total number of codebook embeddings and indicator value of $j^{th}$ codebook vector is sampled for image $x^i$.

$$\text{CBP} = \exp\Big( -\sum_j^K \frac{1}{N} \sum_i^N \text{qidx}_j^i \log \frac{1}{N} \sum_i^N \text{qidx}_j^i \Big) \qquad (11)$$

### D.5    FRECHET INCEPTION DISTANCE (FID)

FID is computed to measure the quality of generated images with respect to real images, it's usually measured by computing the distance between the latent representations of real and generated images. For measuring this we usually use any pre-trained model $\Phi_{vgg}$ for computing latent representation; the FID score is calculated using equation 12, where $z^r = \frac{1}{N} \sum_i^N \Phi_{vgg}(x_i^r); \Sigma^r = \Phi_{vgg}(x^r)^T \Phi_{vgg}(x^r)$ similarly, $z^g$ and $\Sigma^g$ are defined of the generated image set.

$$\text{FID}(x^r, x^g) = ||z^r - z^g||_2^2 + \text{Trace}\Big( \Sigma^r + \Sigma^g - 2\sqrt{(\Sigma^r \Sigma^g)} \Big) \qquad (12)$$

### D.6    SLOT AVERAGE FRECHET INCEPTION DISTANCE (SFID)

As FID measures the quality of the final reconstructed image, to measure the quality of individual slots, we compute FID score with respect original image and individual slot. The intuition behind doing this is if the slot contained object information is present in the original image, the FID for that slot would be less than the slot with random information. We compute the average FID score for all the estimated slots with respect to the original image to obtain SFID, formally described in equation 13

$$\text{SFID}(x^r, x^g) = \frac{1}{K} \sum_i^K \text{FID}(x^r, x_i^g) \qquad (13)$$

### D.7    HUNGARIAN MATCHING COEFFICIENT (HMC)

In the case of the reasoning objective, we measure the implicit rationale provided by the model by comparing the emerging object properties with ground-truth object properties. To consider the permutation invariance property of slot properties, we perform Hungarian Matching over all the properties with respect to the ground truth properties and compute the MSE cost between paired vectors.

# E   ALGORITHM & FORWARD PASS

Our proposed object-level representation learning uses an encoder to map an image to its latent representation, followed by computing $\mathbf{q}, \mathbf{k}, \mathbf{v}$ projection vectors. The obtained position-free encoding $\mathbf{z}$ is further passed through image-level abstraction function $\mathcal{A}$ resulting eigenvectors $\boldsymbol{V}$ and corresponding eigenvalues $\Lambda$ are used to estimate principle components $\mathbf{a}$ as described in section 4. As the obtained principle components are with respect to a particular image, to generalize them across the dataset, GSD sampling is used, $\tilde{\mathbf{k}} \sim \mathfrak{S}^1$. Allowing us to sample slots from uniquely mapped conditioning distributions, ensuring slots to be sampled from the same conditioning distribution $\mathfrak{S}_i^2$ for all instances of an object in the case when the given image has more than one instance of the same object. Based on the heuristic defined in section 4, $K$ slots are sampled from their respective conditioning distributions, which are selected with respect to the obtained eigenvalues. Finally, the iterative attention mechanism is applied to obtain slot-level representations for the downstream tasks. Algorithm 1 describes the proposed algorithm.

# F   QUANTIZATION

In the case of quantization, we first define a codebook $\mathfrak{S} \in \mathbb{R}^{N \times d_z}$ with $N$ codebook embeddings, where individual embedding $\mathfrak{S}_i \in \mathbb{R}^{d_z}$. The objective in the case of quantization is to learn the categorical distribution over codebook features for a given image $\mathbf{x}$. In the case of deterministic sampling, as in the case of Euclidean and Cosine sampling, the categorical distribution is modeled with one-hot probabilities determined by the mapping of each discrete vector to the nearest codebook vector:

$$p(\mathbf{z}_j = \mathfrak{S}_k \mid \mathbf{x}) = \begin{cases} 1 & \text{for } k = \operatorname{argmin}_{\mathfrak{S}_j} \|\mathbf{z} - \mathfrak{S}_j\|^2 \quad \text{or} \quad \langle \mathbf{z}, \mathfrak{S}_j \rangle \\ 0 & \text{otherwise} \end{cases} \tag{14}$$

In this case, the probability estimates are usually non-differentiable. Due to this, in practice, back-propagation over this sampling block is achieved with the help of straight-through gradient approximation, enabling end-to-end training as illustrated in Van Den Oord et al. (2017). Due to the deterministic nature of sampling and the uniform prior assumption, the $D_{\text{KL}}$ term in the objective drops to be constant.

While in the case of stochastic sampling, as in Gumbel, the one-hot probabilities are approximated with the softmax distribution resulting in $p(\mathbf{z} \mid \mathbf{x}) = \texttt{softmax}(\exp((g_i + \hat{z}_i)/t))$, where $g_i$ is sampled from a Gumbel distribution, and $t$ is the temperature parameter as detailed in Maddison et al. (2016); Jang et al. (2016).

# G   CODEBOOK COLLAPSE

Codebook collapse is a common challenge in training vector quantized models, in this case, only one or a selected few embeddings ($>>$ total number of codebook embeddings) of a codebook are being used repeatedly. Various engineering tricks have been proposed to address this issue previously Bardes et al. (2021); Van Den Oord et al. (2017); Träuble et al. (2022); Takida et al. (2022). In this work, we first detach the inputs to the quantization block so that the object-centric representations are not affected by quantizer gradients. We also use the exponential moving average (EMA) over codebook embeddings which encourages the embedding vectors to converge at the mean of the representations. Along with EMA we also utilize random restart of dead codebook vectors with the running representational statistics.

In terms of slot distributions, we initialize the distribution on a unit hypersphere, fix the mean of the distribution, and learn the slot-specific transformation functions which transform the fixed distribution into the required distribution. This was first proposed in Tomczak & Welling (2018) to prevent overfitting of learnable priors.

# H  OBJECT DISCOVERY

Here, we provide more analysis on sensitivity analysis of CoSA on object discovery task. We first test our framework on all three sampling methods using Gumbel, Cosine, and Euclidian codebooks. We use the convolutional encoder and decoder architectures with 5 convolutional and transpose convolutional layers. Additionally, we provide the comparative results on slot properties across different methods.

## H.1  QUANTITATIVE ANALYSIS

Table 6 and 7 demonstrates the sensitivity of CoSA with respect to selected sampling criteria. Based on the results, it can be observed that CoSA performs similarly or better than its slot attention counterpart, illustrating the effectiveness of grounded representations. We measure the diversity in the codebook embeddings and their variation in sampling by measuring CBP and CBD properties, and these properties illustrate the variability in sampling and differences in codebook representations. The resulting values for all five datasets are described in table 8. These results demonstrate that there's no collapse in codebook representations and the sampling (ref. appendix section G for details on codebook collapse). To measure the *goodness* of the generated slots, we measure OPI and SFID which are tabulated in table 9.

Table 6: Sensitivity analysis of CoSA on object discovery.

| METHODS($\downarrow$), METRICS($\rightarrow$) | CLEVR | | TETROMINOES | | OBJECTS-ROOM | | BITMOJI | | FFHQ | |
|---|---|---|---|---|---|---|---|---|---|---|
| | MSE($\downarrow$) | FID($\downarrow$) | MSE($\downarrow$) | FID($\downarrow$) | MSE($\downarrow$) | FID($\downarrow$) | MSE($\downarrow$) | FID($\downarrow$) | MSE($\downarrow$) | FID($\downarrow$) |
| SA | 6.37 | 38.18 | 1.49 | 3.81 | 7.57 | 38.12 | 14.62 | 12.78 | 55.14 | 54.95 |
| CoSA-GUMBEL | 3.82 | 31.84 | **0.21** | **0.26** | 5.45 | 32.41 | 9.84 | 9.36 | 41.77 | 52.21 |
| CoSA-EUCLIDIAN | 4.04 | 33.18 | 1.13 | 1.61 | 6.42 | 34.05 | **8.97** | **8.41** | **31.94** | **35.14** |
| CoSA-COSINE | **3.14** | **29.12** | 0.42 | 0.41 | **4.85** | **28.19** | 8.17 | 9.28 | 33.37 | 36.34 |

## H.2  QUALITATIVE ANALYSIS

Fig. 7(a), 11(a), 13(a), 15(a), and 9(a) illustrate vanilla SA results. Fig. 7(b), 11(b), 13(b), 15(b), and 9(b) illustrate the results of CoSA-Cosine. While, Fig. 8(a), 12(a), 14(a), 16(a), and 10(a) illustrate the results of CoSA-Gumbel and finally, Fig. 8(b), 12(b), 14(b), 16(b), and 10(b) illustrates CoSA-Euclidian results.

## H.3  DINOSAUR ADAPTATION

To extend slot attention to real-world images, we adopt DINOSAUR model Seitzer et al. (2022), which uses the Vision Transformer (ViT-S16) model trained with DINO training strategyCaron et al. (2021). The main objective of DINOSAUR is to reconstruct latent representation rather than reconstructing the original image. To compare our framework under similar specifications, we use DINO as a feature extractor and use CoSA for the extracted latent features to reconstruct these features and use the resulting attention maps as slots. We demonstrate the results of DINOSAUR and variants of DINO-CoSA in Figures 17, 18. Table 1 illustrates the quantitative results on COCO dataset; note that the results in the original work are based

Table 7: ARI on CLEVR6, Tetrominoes, ObjectsRoom, and COCO datasets, for SA baseline model and CoSA-cosine variant. In the case of COCO we use DINOSAUR variant of SA Seitzer et al. (2022) (ViT-S16) as a baseline and use DINO feature extractor (ViT-S16) for CoSA.

| METHOD | TETROMINOES | CLEVR6 | OBJECTSROOM | COCO |
|---|---|---|---|---|
| SA/DINOSAUR | $0.99 \pm 0.005$ | $0.93 \pm 0.002$ | $0.78 \pm 0.02$ | $0.28 \pm 0.02$ |
| CoSA-EUCLIDIAN | $0.99 \pm 0.002$ | $0.94 \pm 0.002$ | $0.81 \pm 0.01$ | $0.27 \pm 0.02$ |
| CoSA-GUMBEL | $0.99 \pm 0.001$ | $0.93 \pm 0.002$ | $0.80 \pm 0.01$ | $0.30 \pm 0.02$ |
| CoSA-COSINE | $0.99 \pm 0.001$ | $\mathbf{0.96 \pm 0.002}$ | $\mathbf{0.83 \pm 0.002}$ | $\mathbf{0.36 \pm 0.01}$ |

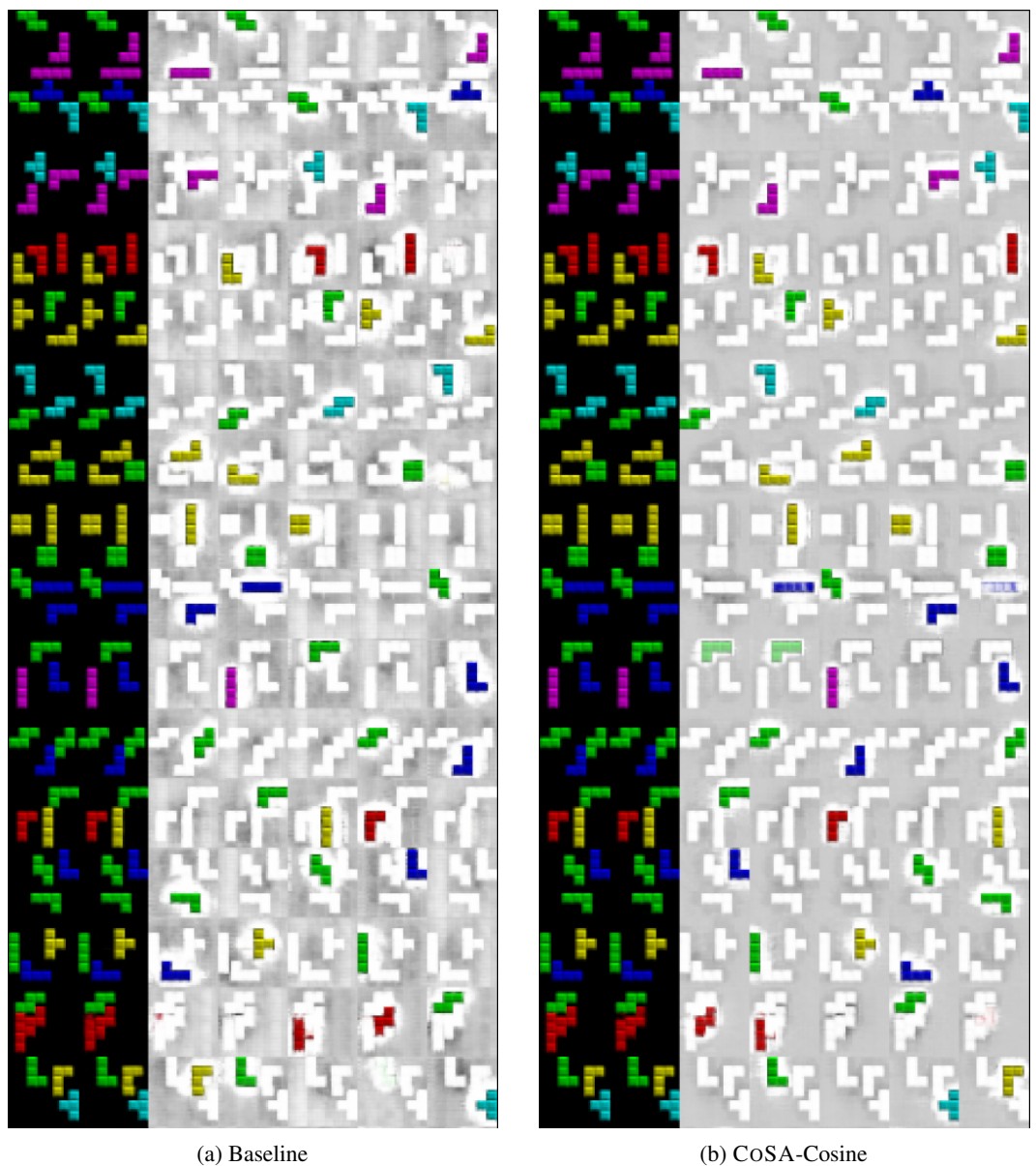

(a) Baseline

(b) COSA-Cosine

Figure 7: Vanilla SA and COSA-Cosine object discovery results on tetrominoes dataset.

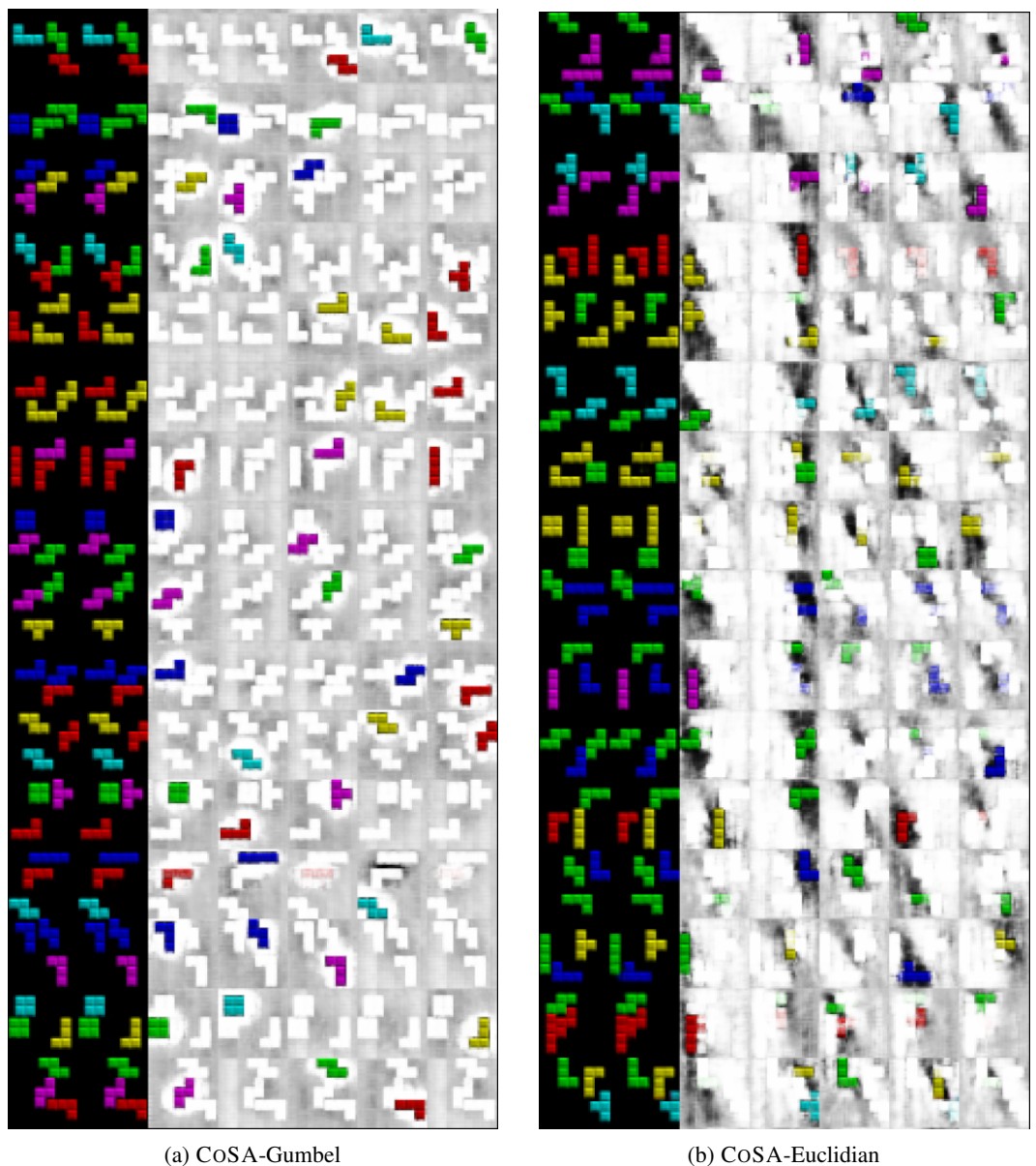

(a) COSA-Gumbel          (b) COSA-Euclidian

Figure 8: COSA-Euclidian and COSA-Gumbel object discovery results on tetrominoes dataset.

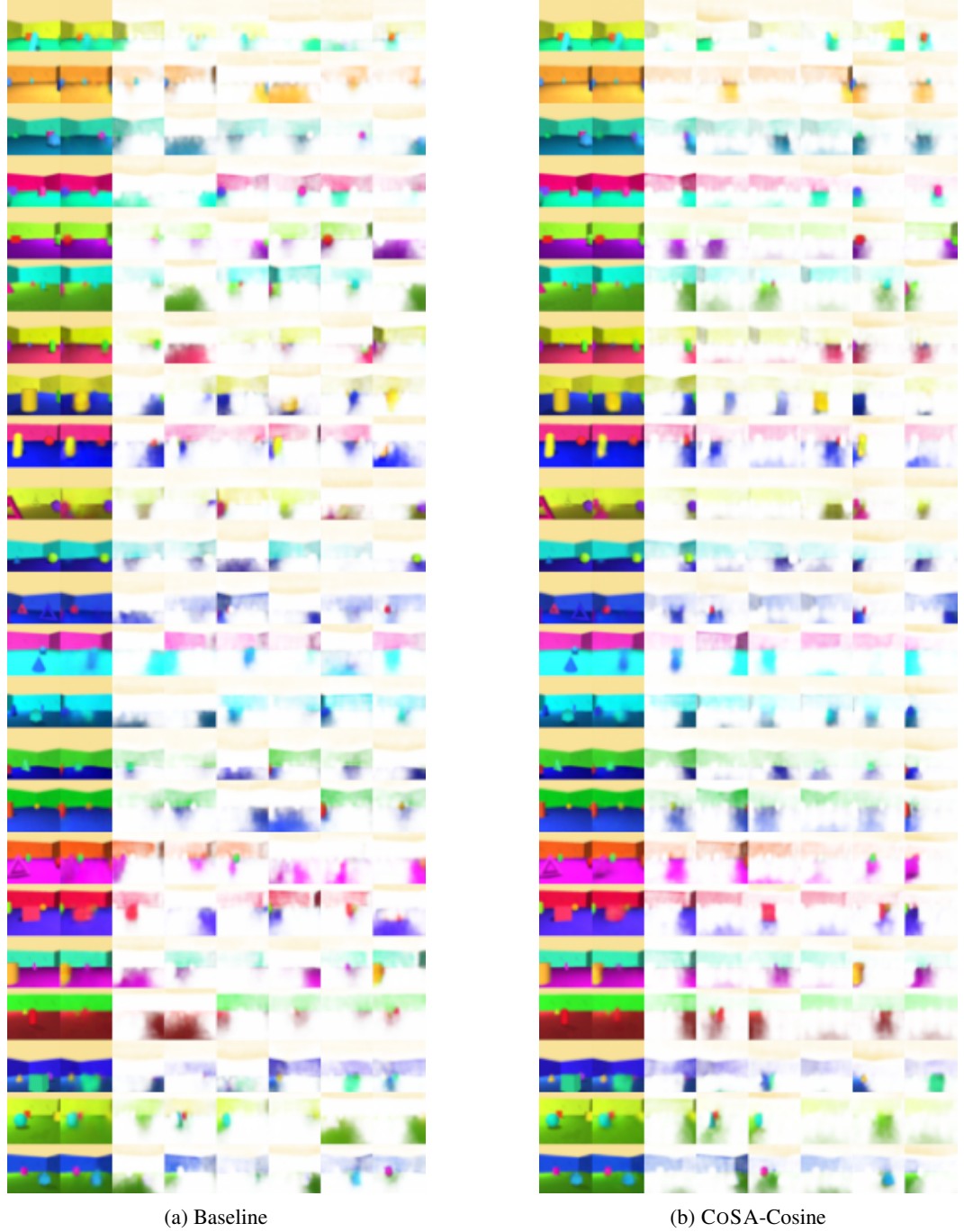

(a) Baseline                                                    (b) COSA-Cosine

Figure 9: Vanilla SA and COSA-Cosine object discovery results on objects-room dataset.

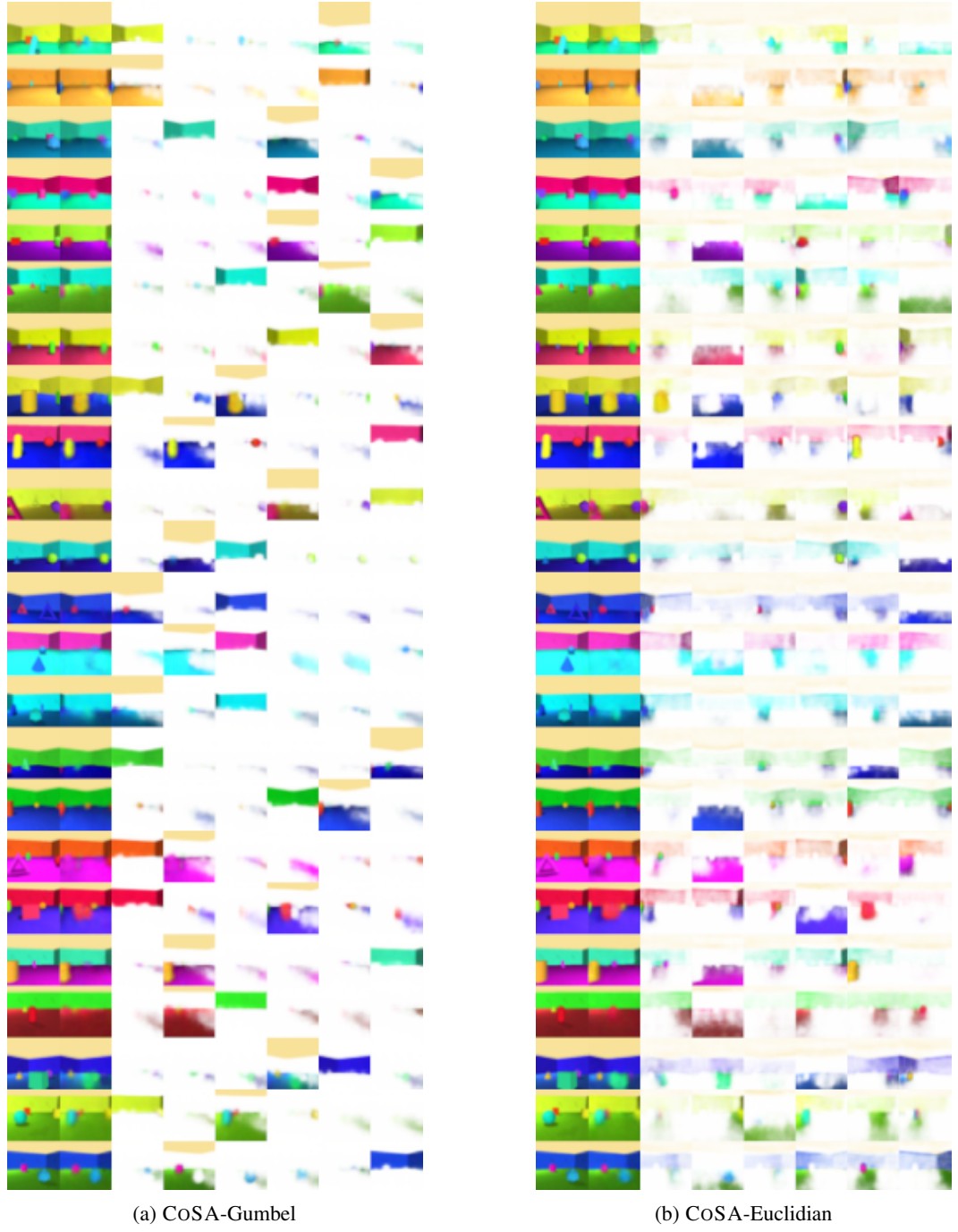

(a) COSA-Gumbel                      (b) COSA-Euclidian

Figure 10: COSA-Euclidian and COSA-Gumbel object discovery results on objects-room dataset.

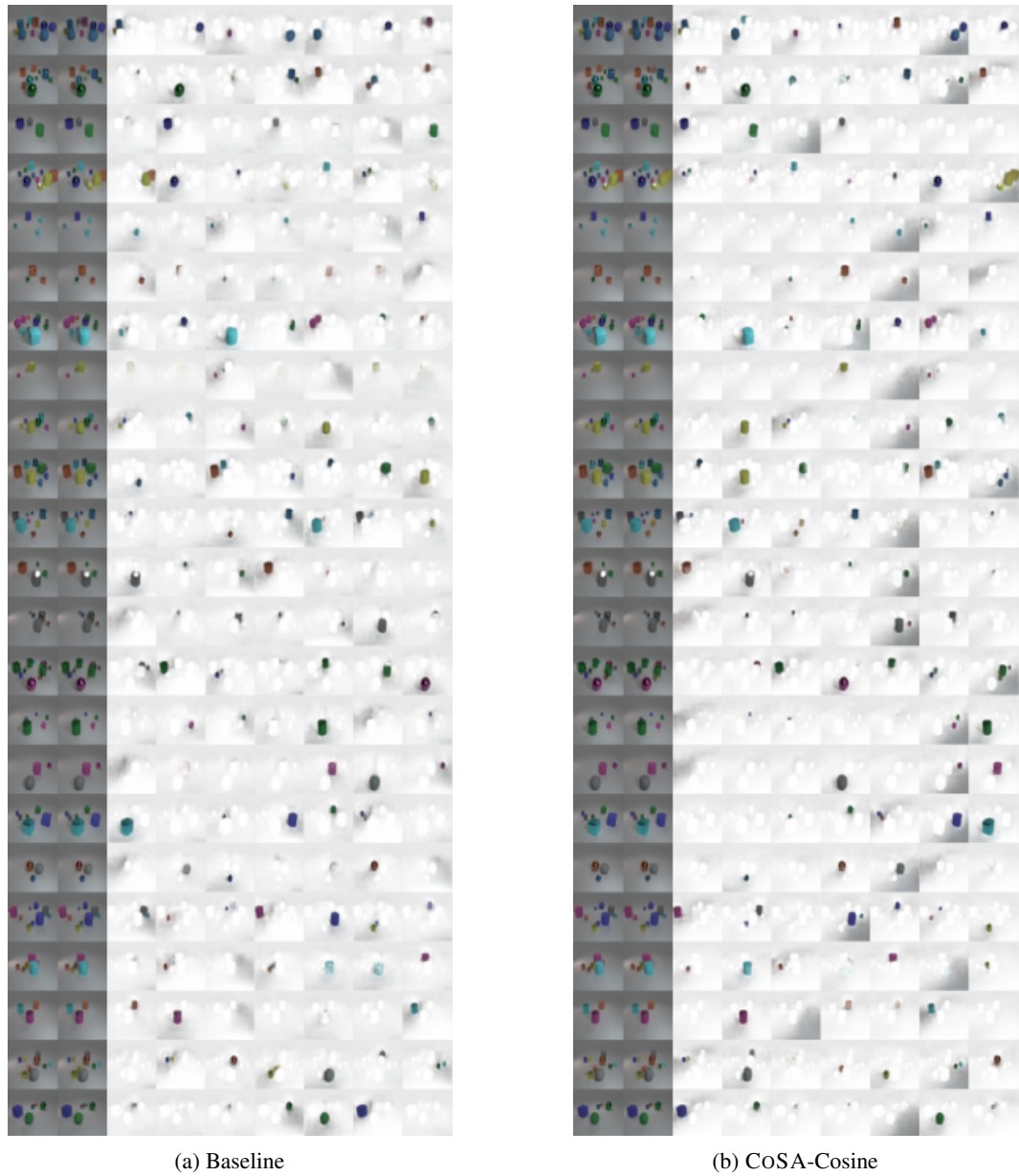

(a) Baseline                    (b) COSA-Cosine

Figure 11: Vanilla SA and COSA-Cosine object discovery results on CLEVR dataset.

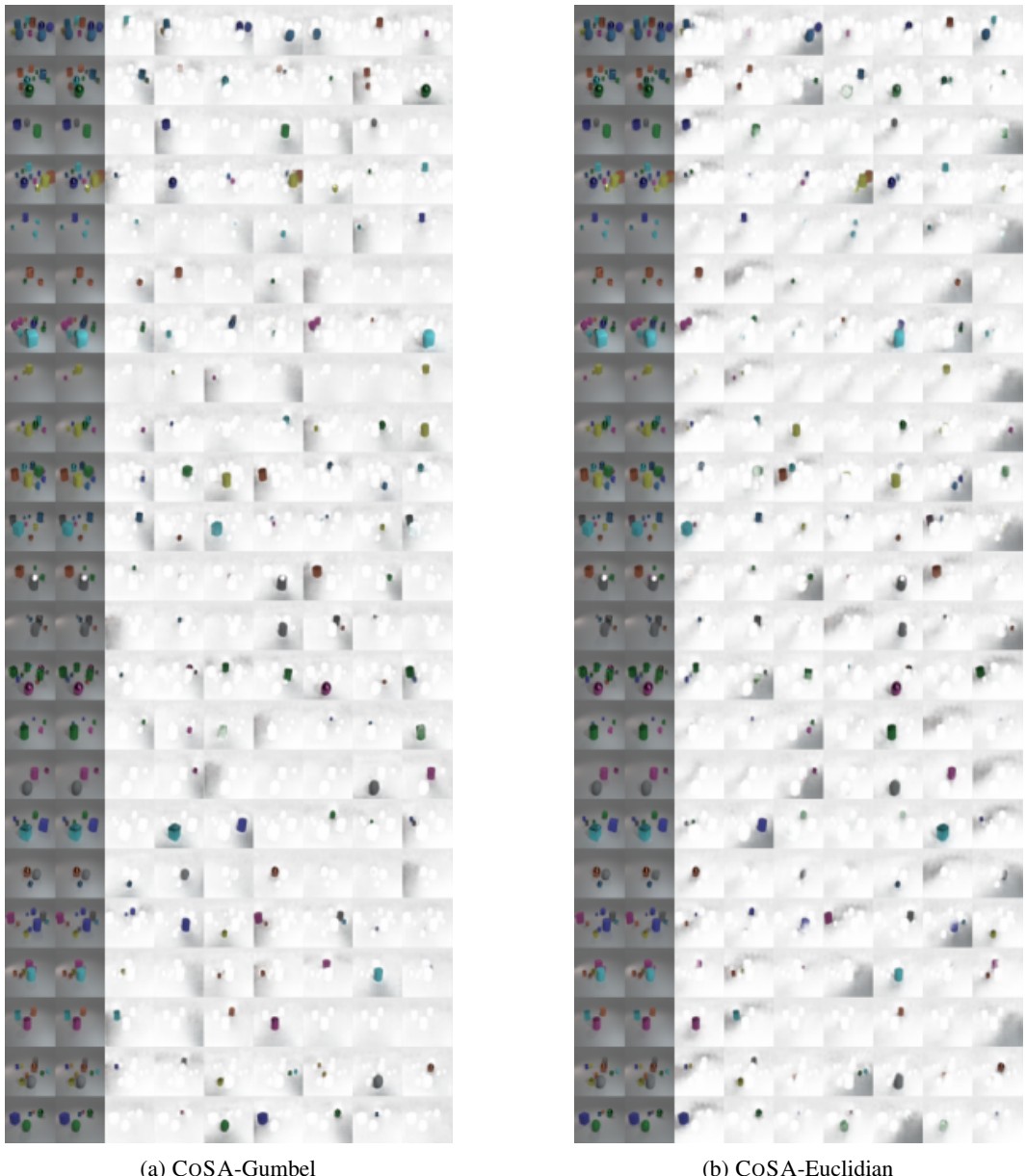

(a) CoSA-Gumbel                                      (b) CoSA-Euclidian

Figure 12: CoSA-Euclidian and CoSA-Gumbel object discovery results on CLEVR dataset.

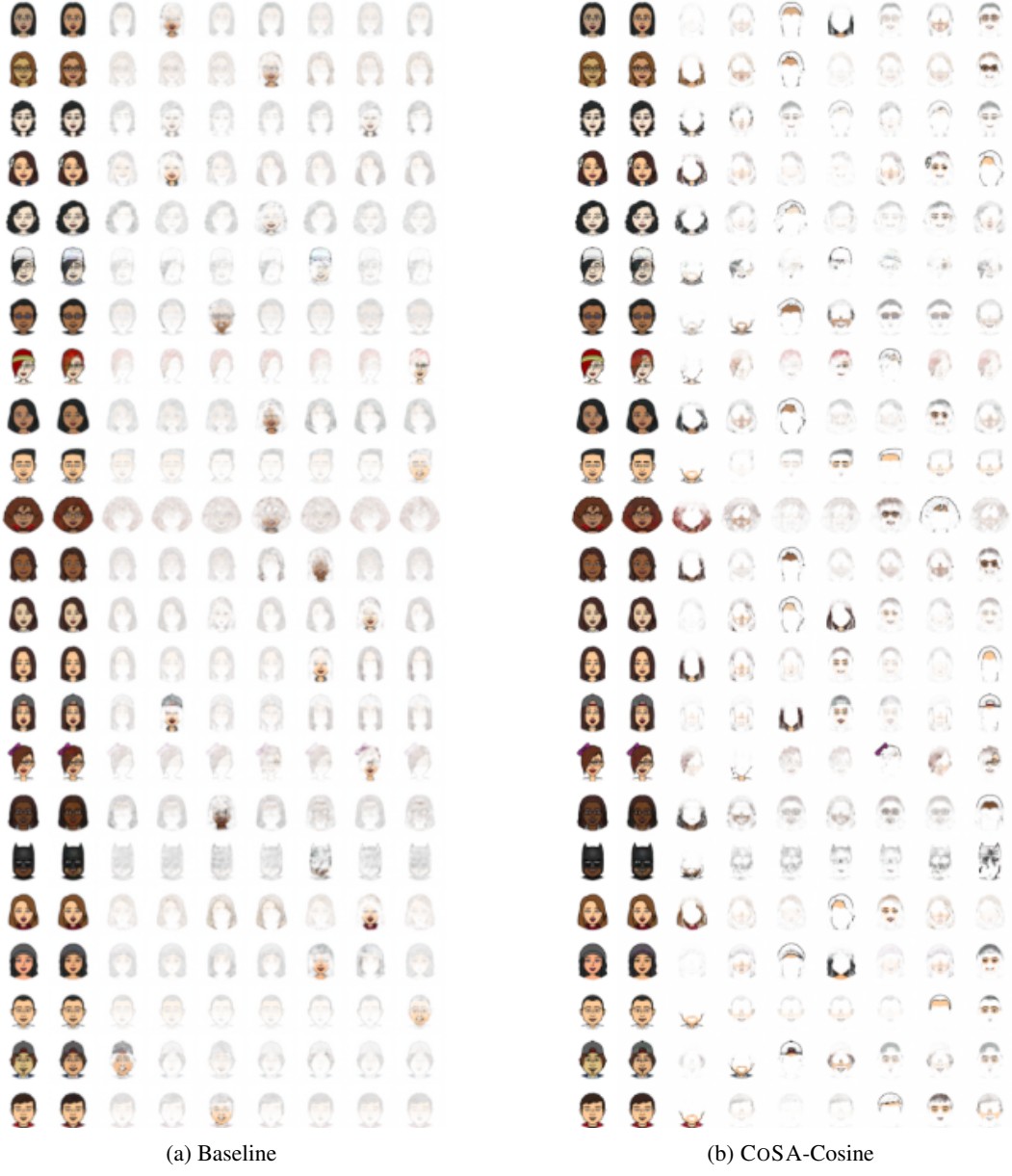

(a) Baseline                    (b) CoSA-Cosine

Figure 13: Vanilla SA and CoSA-Cosine object discovery results on bitmoji dataset.

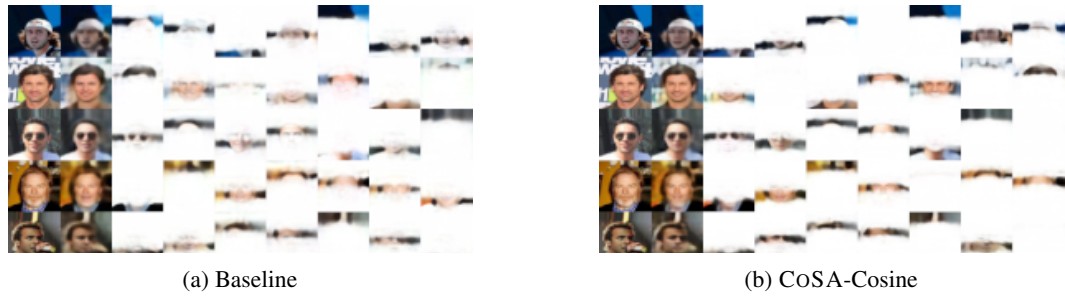

(a) COSA-Gumbel       (b) COSA-Euclidian

Figure 14: COSA-Euclidian and COSA-Gumbel object discovery results on bitmoji dataset.

(a) Baseline       (b) COSA-Cosine

Figure 15: Vanilla SA and COSA-Cosine object discovery results on ffhq dataset.

Table 8: Codebook properties across different sampling methods across datasets.

| METHODS(↓), METRICS(→) | CLEVR | | TETROMINOES | | OBJECTS-ROOM | | BITMOJI | | FFHQ | |
|---|---|---|---|---|---|---|---|---|---|---|
| | CBD(↑) | CBP(↑) | CBD(↑) | CBP(↑) | CBD(↑) | CBP(↑) | CBD(↑) | CBP(↑) | CBD(↑) | CBP(↑) |
| CoSA-COSINE | 0.99 | 44.39 | 0.99 | 26.53 | 0.99 | 28.22 | 0.99 | 54.87 | 0.99 | 36.97 |
| CoSA-EUCLIDIAN | 0.99 | 32.40 | 0.99 | 28.99 | 0.99 | 24.26 | 0.99 | 55.50 | 0.99 | 24.28 |
| CoSA-GUMBEL | 1.0 | 29.62 | 1.0 | 22.35 | 1.0 | 23.05 | 1.0 | 37.61 | 1.0 | 24.69 |

Table 9: Slot properties.

| METHODS(↓), METRICS(→) | CLEVR | | TETROMINOES | | OBJECTS-ROOM | | BITMOJI | | FFHQ | |
|---|---|---|---|---|---|---|---|---|---|---|
| | OPI(↓) | SFID(↓) | OPI(↓) | SFID(↓) | OPI(↓) | SFID(↓) | OPI(↓) | SFID(↓) | OPI(↓) | SFID(↓) |
| SA | 0.43 | 238.7 | 0.48 | 270.3 | 0.42 | 162.9 | 0.41 | 204.1 | 0.27 | 240.9 |
| IMPLICIT | **0.35** | 220.1 | 0.56 | 249.1 | 0.37 | 128.5 | 0.34 | 197.4 | 0.24 | 244.1 |
| CoSA-GUMBEL | **0.35** | 192.7 | 0.43 | **231.2** | 0.42 | 119.4 | 0.30 | 201.2 | 0.28 | 213.1 |
| CoSA-EUCLIDIAN | 0.40 | **173.1** | **0.34** | 236.7 | **0.34** | 137.5 | **0.29** | **190.3** | 0.33 | 232.3 |
| CoSA-COSINE | 0.45 | 175.8 | 0.53 | 235.2 | 0.39 | **103.2** | 0.37 | 213.3 | **0.32** | **211.3** |

on the ViT-B16 feature extractor, while we use ViT-S16 due to computational resources. We base our implementation by considering base implementation from `https://github.com/amazon-science/object-centric-learning-framework` and use provided hyperparameters in 'coco_feat_rec_in_small16_auto.yaml', with only changes in the number of GPUs and the batch size; we use a single GPU with a batch size of 32.

## H.4 CODEBOOK SIZE ANALYSIS

Given we make dictionary sufficiency assumption (refer to an assumption 4), we perform extensive ablations on dictionary size on CLEVR data for all three sampling strategies. Table 10, 11, and 12 correspond to Cosine, Euclidean, and Gumbel sampling strategies. As it can be observed, a very low number of codebook embeddings (M=16) results in high MSE error and lower ARI scores, while increasing the embeddings drastically also does not help. While the number of embeddings around 64, 128 provide similar results without any collapse (indicated by higher perplexity scores). This indicates that the framework is not extremely sensitive to the codebook size, given the codebook does not collapse.

Table 10: Codebook size ablations with Cosine Sampling

| METHODS(↓), METRICS(→) | CLEVR | | | |
|---|---|---|---|---|
| | MSE(↓) | FG-ARI(↑) | CBD(↑) | CBP(↑) |
| M=16 | 8.11 | 0.73 | 0.99 | 14.96 |
| M=64 | 3.14 | 0.96 | 0.99 | 44.39 |
| M=128 | 3.46 | 0.94 | 0.98 | 51.65 |
| M=256 | 6.68 | 0.85 | 0.98 | 82.20 |

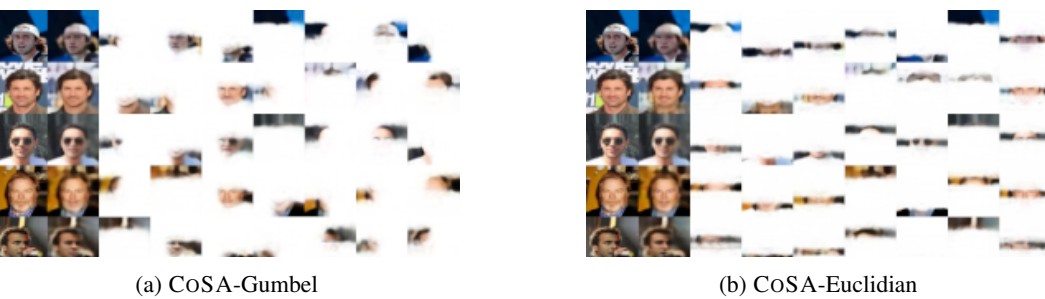

(a) CoSA-Gumbel

(b) CoSA-Euclidian

Figure 16: CoSA-Euclidian and CoSA-Gumbel object discovery results on ffhq dataset.

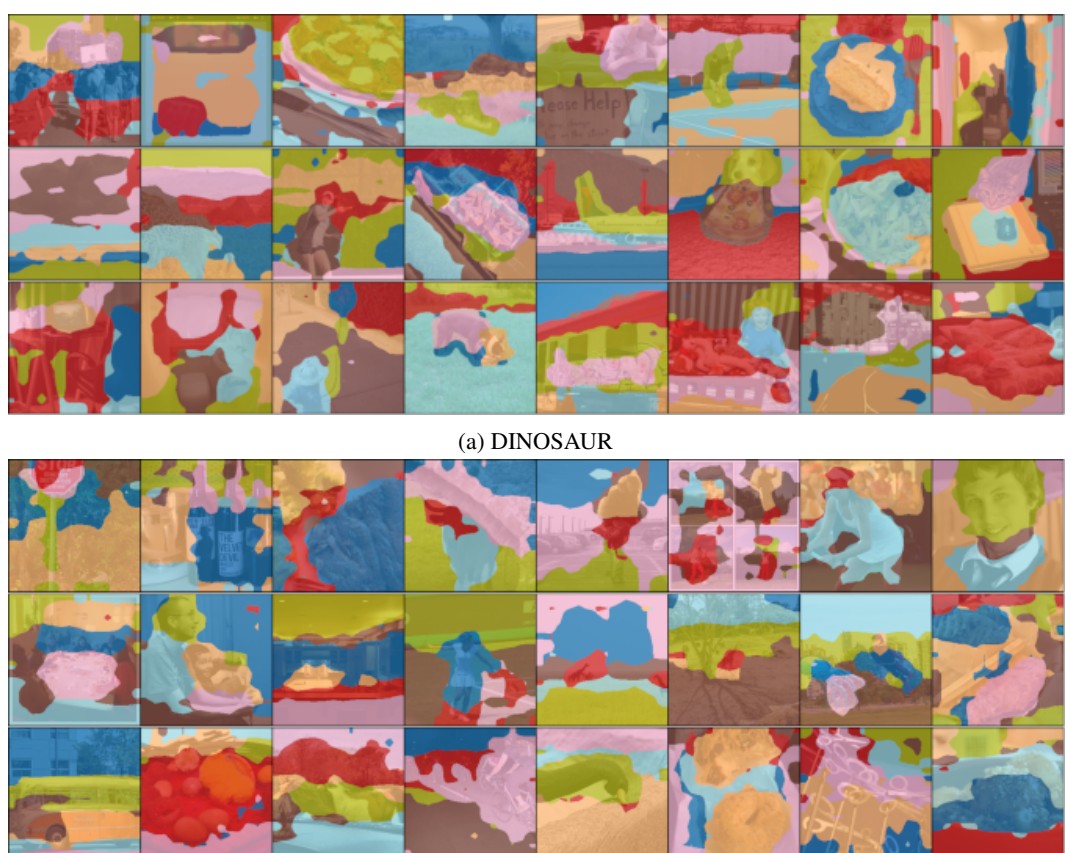

(a) DINOSAUR

(b) DINO-CoSA-Euclidian

Figure 17: Results on COCO dataset.

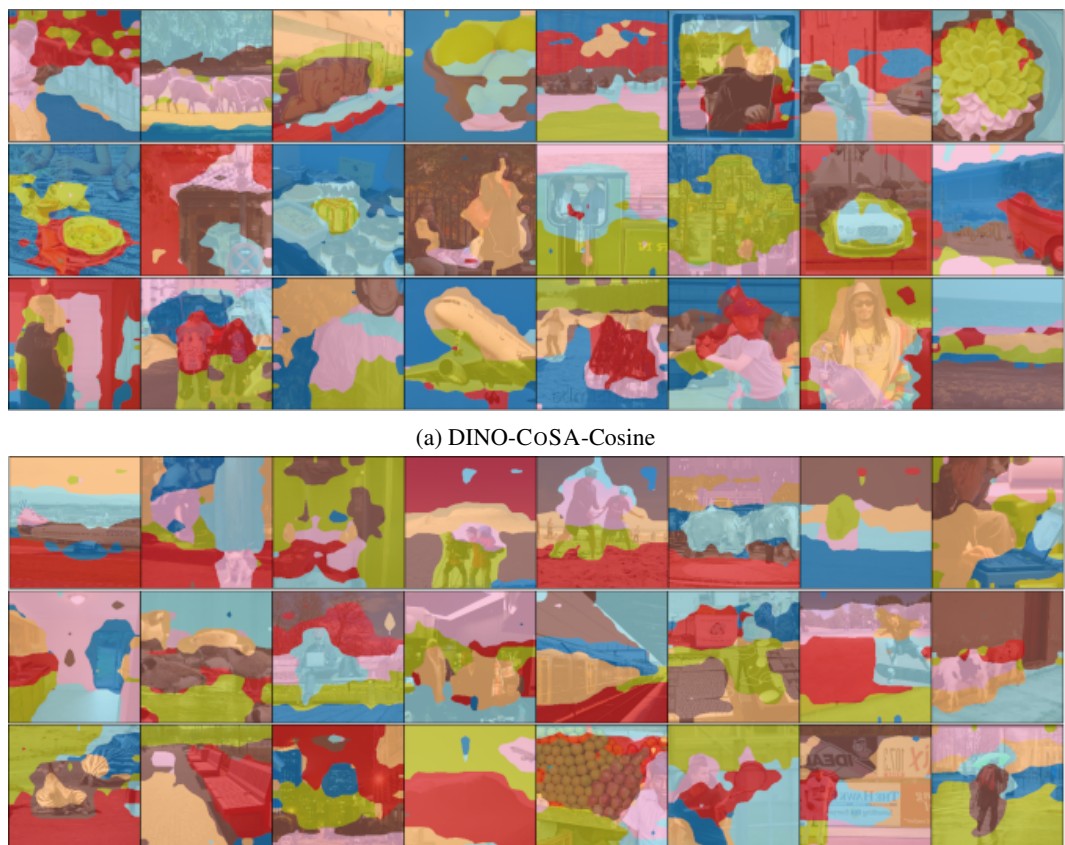

(a) DINO-CoSA-Cosine

(b) DINO-CoSA-Gumbel

Figure 18: Results on COCO dataset.

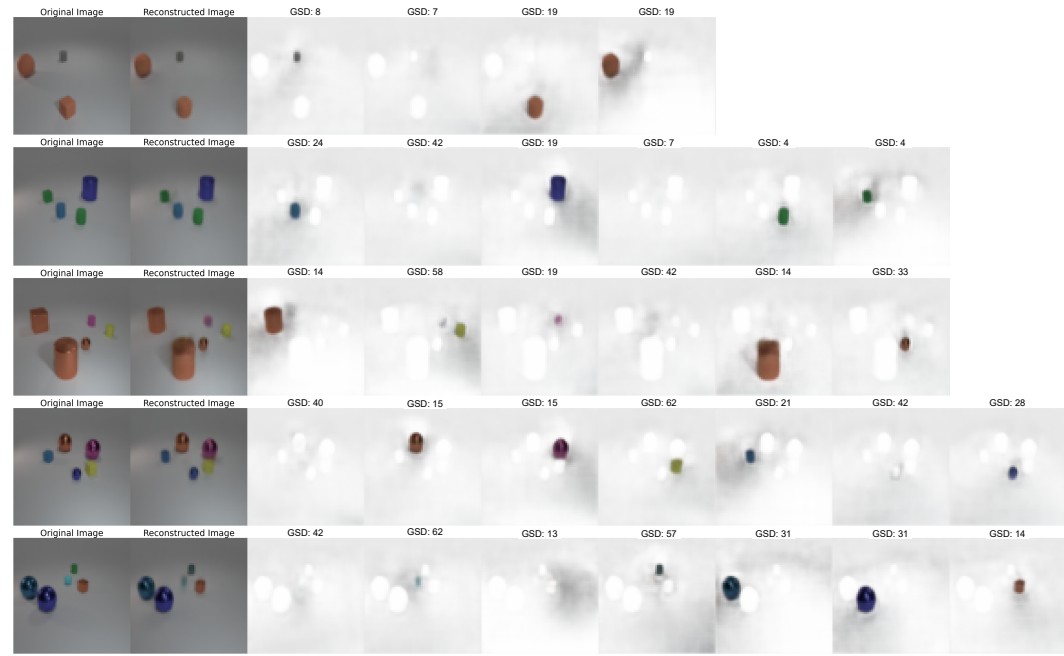

Figure 19: Demonstration of object binding across instances in a given image. The title of each figure corresponds to the sampled element index of GSD $\mathfrak{S}$.

Table 11: Codebook size ablations with Euclidian Sampling

| METHODS($\downarrow$), METRICS($\rightarrow$) | CLEVR | | | |
|---|---|---|---|---|
| | MSE($\downarrow$) | FG-ARI($\uparrow$) | CBD($\uparrow$) | CBP($\uparrow$) |
| M=16 | 4.81 | 0.93 | 0.99 | 8.33 |
| M=64 | 4.04 | 0.94 | 0.99 | 32.40 |
| M=128 | 3.64 | 0.94 | 0.95 | 20.69 |
| M=256 | 4.34 | 0.89 | 0.96 | 19.79 |

Table 12: Codebook size ablations with Gumbel Sampling

| METHODS($\downarrow$), METRICS($\rightarrow$) | CLEVR | | | |
|---|---|---|---|---|
| | MSE($\downarrow$) | FG-ARI($\uparrow$) | CBD($\uparrow$) | CBP($\uparrow$) |
| M=16 | 6.64 | 0.88 | 1.0 | 11.06 |
| M=64 | 3.82 | 0.93 | 1.0 | 29.62 |
| M=128 | 3.89 | 0.93 | 0.99 | 27.16 |
| M=256 | 5.46 | 0.92 | 0.99 | 30.28 |

## H.5 MULTIPLE OBJECT INSTANCES

In this section, we demonstrate the effectiveness of COSA on binding multiple instances of an object in a given image to a same element in GSD. In Figure 19, the title of each sub figure correspond to GSD index being selected for that particular object. As observed in first-fifth row GSD elements $\mathfrak{S}_{19}, \mathfrak{S}_4, \mathfrak{S}_{14}, \mathfrak{S}_{15}$, and $\mathfrak{S}_{31}$ are being sampled twice, respectively.

Similarly, to previous analysis we use the coco trained model and analyse the binding capabilities across multiple frames in a video sequence. For this we use the COCO trained model and perform zero-shot adaptation in DAVIS video dataset Perazzi et al. (2016). Figure 20 reflects this analysis,

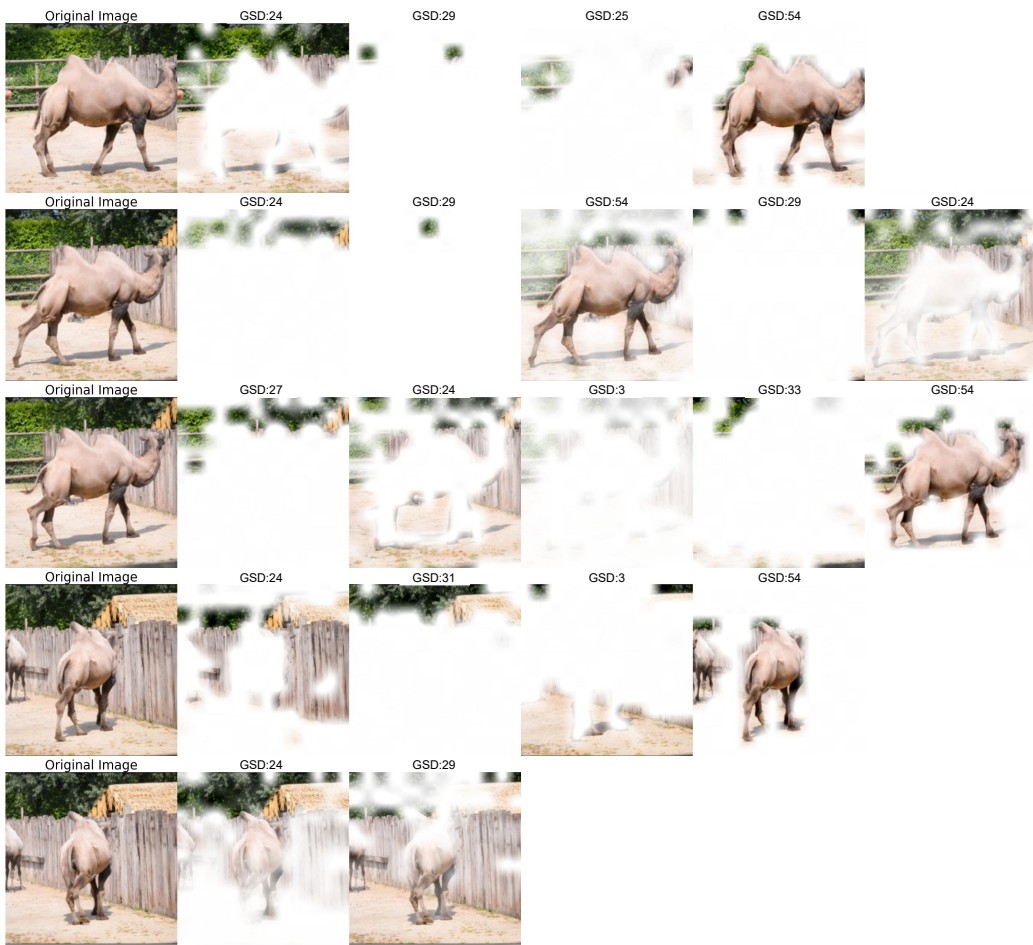

Figure 20: Demonstration of object binding across frames in a video sequence. The title of each figure corresponds to the sampled element index of GSD $\mathfrak{S}$.

where each row correspond to a particular frame in a vide. We can observed that GSD elements $\mathfrak{S}_{54}$, and $\mathfrak{S}_{24}$ consistently map to camel and the background, respectively.

# I   ABSTRACTION FUNCTION

We tried multiple variants of spectral decomposition for abstracting object-level features including singular value decomposition, QR decomposition, vanilla quantization, and eigenvalue decomposition. Table 13 describes the results on the CLEVR dataset for all different types of variants of abstraction functions. We use the principle components based decomposition approach as the default abstraction function because the results do not vary too much among different types of spectral decomposition methods.

Table 13: Abstraction functions

| METHODS(↓), METRICS(→) | VANILLA QUANT. | | PRINCIPLE VEC. QUANT. | | QR VEC. QUANT. | | SINGULAR VEC. QUANT. | |
|---|---|---|---|---|---|---|---|---|
| | MSE(↓) | FID(↓) | MSE(↓) | FID(↓) | MSE(↓) | FID(↓) | MSE(↓) | FID (↓) |
| CoSA-GUMBEL | 6.44 | 43.61 | 3.82 | 29.12 | 4.18 | 33.74 | 3.97 | 26.29 |
| CoSA-EUCLIDIAN | 8.32 | 54.11 | 4.04 | 33.18 | 4.76 | 32.43 | 4.14 | 28.57 |
| CoSA-COSINE | 7.68 | 48.72 | 3.14 | 31.84 | 3.64 | 34.12 | 3.98 | 30.22 |

## J CONVERGENCE OF COSA

**Proposition 3.** *Under assumptions 1-5 the solution $(\alpha, \beta, \gamma, \theta, \phi, \psi)$ satisfies $\mathbb{E}_{\mathbf{s}^0 \sim p(\mathbf{s}^0 | \tilde{\mathbf{z}})}(\mathbf{s}^0) = \mathbb{E}_{\tilde{\mathbf{s}} \sim p(\tilde{\mathbf{s}} | \mathbf{x})}(\tilde{\mathbf{s}})$, where $\tilde{\mathbf{s}}$ is a representation of a true object, resulting in the convergence of the learned canonical representation to mean of true slot distrbutions.*

*Proof.* For proof, we make use of the slot posterior described in Equation 3

$$p(\mathbf{s}^T \mid \mathbf{x}) = \delta \left( \mathbf{s}^T - \prod_{t=1}^{T} \mathcal{H}_\theta \left( \mathbf{s}^{t-1}, f(g(\hat{\mathbf{q}}^{t-1}, \mathbf{k}), \mathbf{v}) \right) \right)$$

where $\delta(\cdot)$ is Dirac delta distributed given randomly sampled initial slots from their marginals $\mathbf{s}^0 \sim p(\mathbf{s}^0 \mid \tilde{\mathbf{z}}) = \prod_{i=1}^{K} \mathcal{N}\left(\mathbf{s}_i^0; \boldsymbol{\mu}_i, \boldsymbol{\sigma}_i^2\right)$, associated with the codebook vectors $\tilde{\mathbf{z}} \sim q(\tilde{\mathbf{z}} \mid \mathbf{x})$. The distribution over the initial slots $\mathbf{s}^0$ induces a distribution over the refined slots $\mathbf{s}^T$.

Let $\omega = \{\alpha, \beta, \gamma, \theta, \psi, \phi\}$ set of all the parameters. Then, the joint distribution for all slots can be described as:

$$p_\omega(\mathbf{s}^T, \mathbf{s}^0, \tilde{\mathbf{z}} \mid \mathbf{x}) = \delta \left( \mathbf{s}^T - \prod_{t=1}^{T} \mathcal{H}_\theta \left( \mathbf{s}^{t-1}, f(g(\hat{\mathbf{q}}^{t-1}, \mathbf{k}), \mathbf{v}) \right) \right) q^i(\tilde{\mathbf{z}} \mid \mathbf{x}) p(\mathbf{s}^0 \mid \tilde{\mathbf{z}})$$

By assumptions 2, 3, and 4, we know that the slot distribution is structurally identifiable. Structural identifiability property of the model is explored in detail by Brady et al. (2023).

Given structural identifiability of the model, to show $\mathbb{E}_{\mathbf{s}^0 \sim p(\mathbf{s}^0 | \tilde{\mathbf{z}})}(\mathbf{s}^0) = \mathbb{E}_{\tilde{\mathbf{s}} \sim p(\tilde{\mathbf{s}} | \mathbf{x})}(\tilde{\mathbf{s}})$, under infinite data regime, we need to show that the likelihood estimate follows:

$$\lim_{N \to \infty} \frac{1}{N} \sum_{k=0}^{N} \log p_\omega(\mathbf{s}_k^0 \mid \mathbf{x}_k) \leq \lim_{N \to \infty} \frac{1}{N} \sum_{k=0}^{N} \log p(\tilde{\mathbf{s}}_k \mid \mathbf{x})$$

where $\mathbf{s}_k$ corresponds to all the slots estimated for a datapoint $\mathbf{x}_k$.

By the law of large numbers, we know that:

$$\lim_{N \to \infty} \frac{1}{N} \sum_{k=0}^{N} \log p_\omega(\mathbf{s}_k^0 \mid \mathbf{x}_k) = \mathbb{E}_{p(\tilde{\mathbf{s}}, \mathbf{x})}[\log p_\omega(\mathbf{s}^0 \mid \mathbf{x})]$$

Now we can show:

$$\mathbb{E}_{p(\tilde{\mathbf{s}}, \mathbf{x})}[\log p_\omega(\mathbf{s}^0 \mid \mathbf{x})] - \mathbb{E}_{p(\tilde{\mathbf{s}}, \mathbf{x})}[\log p(\tilde{\mathbf{s}}_i \mid \mathbf{x})]$$

$$= \mathbb{E}_{p(\tilde{\mathbf{s}}, \mathbf{x})} \left[ \log \frac{p_\omega(\mathbf{s}^0 \mid \mathbf{z} = \Phi_e(\mathbf{x})))}{p(\tilde{\mathbf{s}} \mid \mathbf{x})} \right]$$

$$\leq \mathbb{E}_{p(\tilde{\mathbf{s}}, \mathbf{x})} \left[ \frac{p_\omega(\mathbf{s}^0 \mid \mathbf{z} = \mathbf{x})))}{p(\tilde{\mathbf{s}} \mid \mathbf{x})} - 1 \right] \qquad (\because \quad \log t \leq t - 1)$$

$$= \int p_\omega(\mathbf{s}^0 \mid \mathbf{z} = \mathbf{x}) d\mathbf{s} - 1 = 0$$

$\square$

**Discussion.** This mainly indicates that the sampled from $p(\mathbf{s}^0 \mid \tilde{\mathbf{z}})$ are closer to the mean of the tru distribution, given infinite data and under specified assumptions ELBO estimate can recover this slot distribution.

**Lemma 1.** *(Convergence rate) Given the per iteration update in SA, $\mathbf{u}^1 = f(g(\mathbf{k}, \hat{\mathbf{q}}), \mathbf{v})$ and per iteration update of COSA $\mathbf{u}^2 = f(g(\tilde{\mathbf{k}}, \hat{\mathbf{q}}), \mathbf{v})$, there exists $\Delta \geq 0$ such that $\hat{d}(\mathbf{u}^1, \tilde{\mathbf{s}}) - \hat{d}(\mathbf{u}^2, \tilde{\mathbf{s}}) = \Delta$ with $\tilde{\mathbf{s}}$ being representation of true objects.*

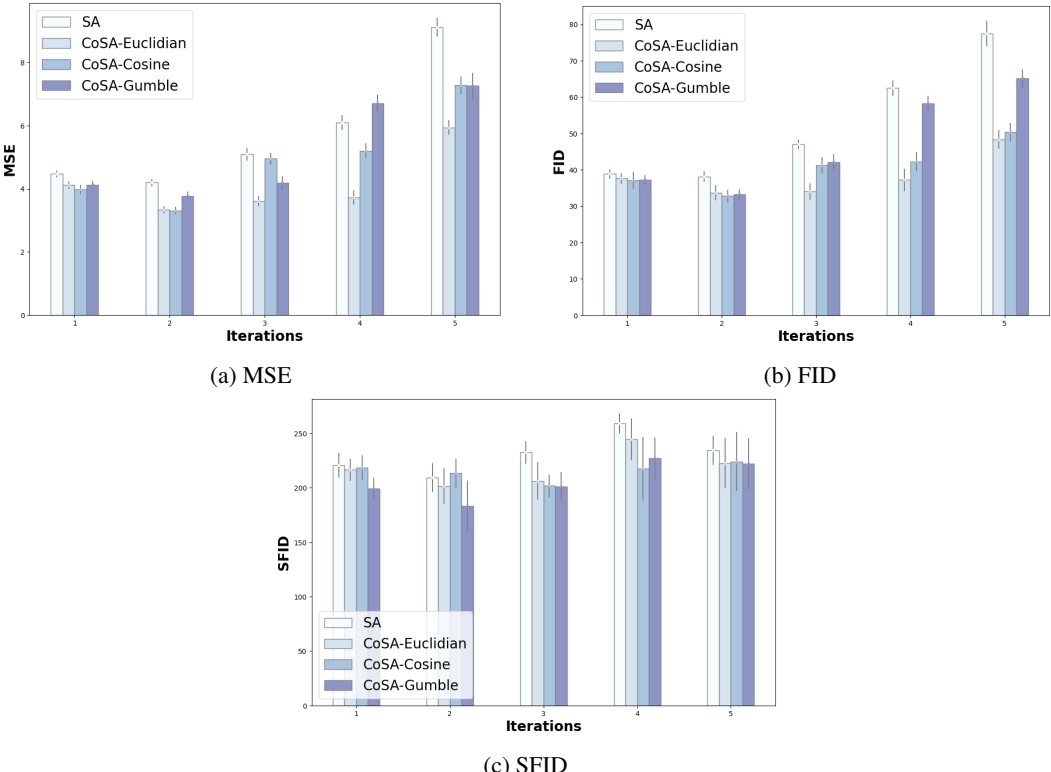

(a) MSE                                                      (b) FID

(c) SFID

Figure 21: Iteration wise property comparison between methods with full-backpropagation without Neumann's approximation for CLEVR dataset.

*Remark* 5. In practice, the parameters in the update function and query, key, and value projection functions differ in both cases. Here, we are interested in investigating the effect of initializing slots concerning specific conditionals rather than using joint distribution.

*Proof.* The proof directly follows from the result of a proposition 3 , and we include the proof for completion.

For a given the result of a proposition 3 , we know that $\hat{d}(\mathbf{s}^0, \tilde{\mathbf{s}}) - \hat{d}(\bar{\mathbf{s}}^0, \tilde{\mathbf{s}}) \geq 0$, where $\mathbf{s}^0, \bar{\mathbf{s}}^0$ correspond to initial slots sampled in SA and COSA respectively and $\hat{d}$ is a convex distance functions.

Updates at iteration 0, $\mathbf{u}_0^1 = f(g(\mathbf{k}, \mathcal{Q}(\mathbf{s}^0)), \mathbf{v}), \mathbf{u}_0^2 = f(g(\mathbf{k}, \mathcal{Q}(\bar{\mathbf{s}}^0)), v)$, since $f(.)$ is convex combination function over elements of $v$ and with weights obtained from an attention map $g(.)$. The composite function is monotonic, resulting in $\hat{d}(\mathbf{u}_0^1, \tilde{\mathbf{s}}) - \hat{d}(\mathbf{u}_0^2, \tilde{\mathbf{s}}) \geq 0$.

Similarly, $h$ is also a monotonic function with matrix product and $tanh$ activation, resulting in a general expression $\hat{d}(\mathbf{u}^1, \tilde{\mathbf{s}}) - \hat{d}(\mathbf{u}^2, \tilde{\mathbf{s}}) \geq 0$.

□

The easier way to understand the convergence rate in COSA is that, here the slot initialization is based on the grounded (mean) object representation, rather than random, which requires fewer iterations to converge to the final object. To illustrate the correctness of Lemma 1, we consider the object discovery task on the CLEVR dataset and compare results in terms of MSE, FID, and SFID metrics by varying numbers of iterations in SA and COSA variants. Here, we compute the properties without Neumann's approximation (*i.e.,* without gradient truncation) to compare the vanilla version of SA and COSA. Fig. 21 illustrates the faster convergence, validating our lemma.

---

**Algorithm 2** Scene composition by random slot prompting

---

1: **Prompt Dictionary Items:** index $\sim \mathcal{U}\{0, M\}$
2: **Sample Inputs:** $\mathbf{z} = \mathfrak{S}^1[\text{index}]$
3: **Compute:** $\mathbf{k} = \mathcal{K}_\beta(\mathbf{z})$, and $\mathbf{v} = \mathcal{V}_\phi(\mathbf{z})$
4: **for** $i = 0, \ldots, R$              ▷ $R$ Monte Carlo samples
5:      $\text{slots}_i^0 \sim \mathfrak{S}[\text{index}] \in \mathbb{R}^{K \times d_s}$       ▷ Sample prompted slots from GSD
6:      **for** $t = 1, \ldots, T$          ▷ Refine slots over $T$ attention iterations
7:         $\text{slots}_i^t = f\left(g\left(\mathcal{Q}_\gamma\left(\text{LayerNorm}\left(\text{slots}_i^{t-1}\right)\right), \mathbf{k}\right), \mathbf{v}\right)$    ▷ Update slot representations
8:         $\text{slots}_i^t \mathrel{+}= \text{MLP}\left(\text{LayerNorm}\left(\mathcal{H}_\theta\left(\text{slots}_i^{t-1}, \text{slots}_i^t\right)\right)\right)$    ▷ GRU update & skip connection
9: **return** $\sum_i \text{slots}_i / R$           ▷ MC estimate

---

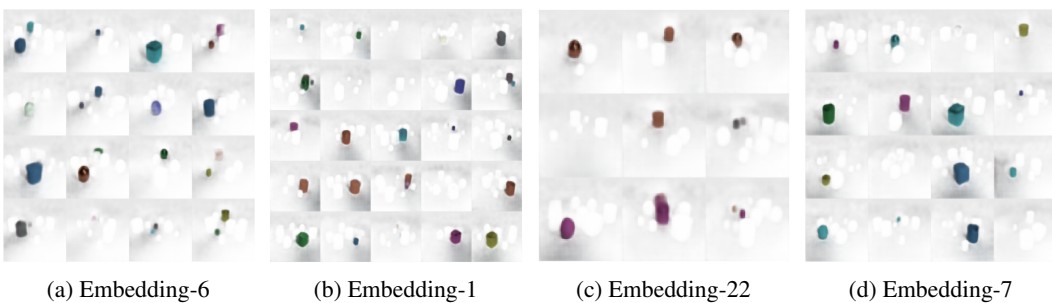

    (a) Embedding-6        (b) Embedding-1        (c) Embedding-22       (d) Embedding-7

Figure 22: Gumbel dictionary embedding to observation mapping for CLEVR dataset.

## K  OBJECT COMPOSITION

Unlike SLATE Singh et al. (2021), our approach does not require test time prompt generation, as the dictionary behaves as a set of prompts; we do not need to learn prompt dictionary, as we learn object/property level distribution as part of our algorithm. The prompting in our approach is solely based on the learned slot dictionary. Algorithmically, we first sample input features from a dictionary and corresponding slots, followed by positional matting and iterative slot refinement for generating realistic object-level scenes. Algorithm 2 describes the approach with pseudo-code.

Object composition illustrates that the grounded representation (every embeddings in a GSD) are specialised to capture specific individual object in a given environment, as expected in proposition 3. We select $K$ slots, sample an embeddings from the respective slot-distributions and refine them generating $K$ 4 channeled images, which we further combine by merging them using softmax distribution, providing us with the prompted image.

### K.1  SLOT DICTIONARY ANALYSIS

Here, we describe the mapped dictionary elements, Fig. 3, 22, 23, and 24 illustrate randomly selected dictionary index to slot mapping. To generate these images, we list the sampled codebook index for given 100 images and group them with respect to uniquely sampled codebook indices. It can be observed that the codebook kind groups similar-looking object properties together. In Fig. 3, it can be seen that embedding-7 of the codebook corresponds to cheeks, embedding-14 maps to head ware and forehead, embedding-25 maps to eyes, and embedding-55 maps to facial hair. It is important to note that these semantics are completely subjective and assigned by observation. Similar observations can be made on CLEVER and Object-Room models illustrated in Fig. 22 and Fig. 23, respectively. The model still struggles to disentangle in the case of natural images like the FFHQ dataset as illustrated in Fig. 24, nevertheless, it is grouping some similar-looking regions together.

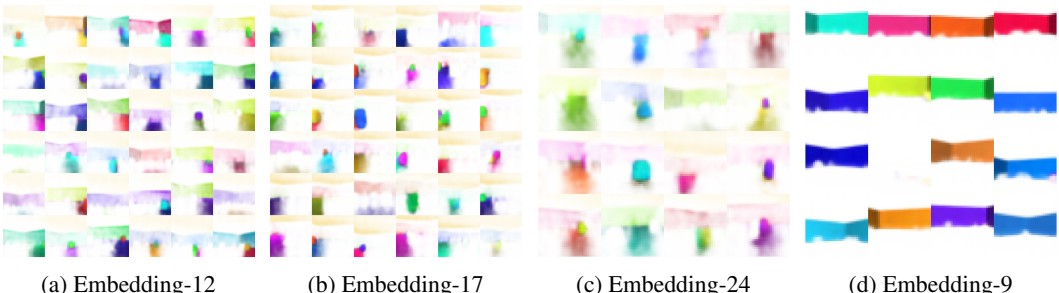

| (a) Embedding-12 | (b) Embedding-17 | (c) Embedding-24 | (d) Embedding-9 |

Figure 23: Cosine dictionary embedding to observation mapping for Objects-room dataset.

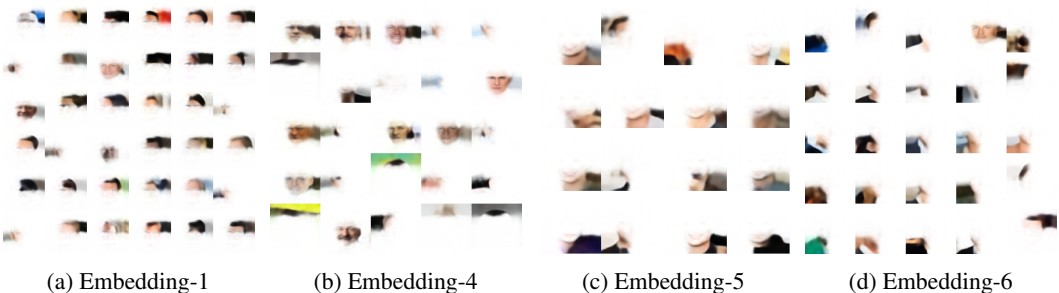

| (a) Embedding-1 | (b) Embedding-4 | (c) Embedding-5 | (d) Embedding-6 |

Figure 24: Gumbel dictionary embedding to observation mapping for FFHQ dataset.

### K.2 SCENE GENERATION

Finally, we also demonstrate the image composition objective by randomly sampling embeddings from the codebook and performing iterative attention to generate results. Fig. 25, 26, and 27 illustrate both randomly sampled slots and the resulting image composition.

## L DISCRIMINATIVE/REASONING TASKS

For validating discriminative tasks, we consider CLEVR-Hans and FloatingMNIST tasks, where the main objective is to segregate the image into different classes based on the contents of the image, while proving a rational for the segregation without explicitly training for the rational. More details about the datasets are described in C. As the classification is based on the object properties in the images, we consider the properties as *emerging* entities and model it as a reasoning task by considering the emerging properties as a rationale for prediction. The correctness of the generated rationale is measured using Hungarian Matching Coefficient (HMC).

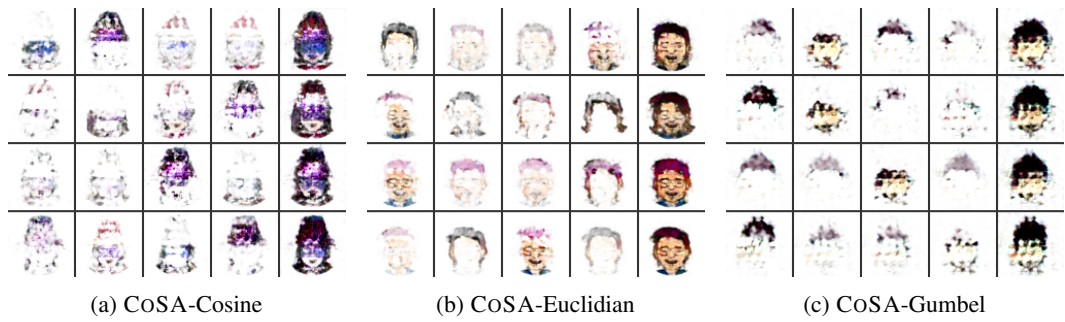

| (a) CoSA-Cosine | (b) CoSA-Euclidian | (c) CoSA-Gumbel |

Figure 25: Image composition with random slot prompt on bitmoji dataset.

Table 14: Compositional FID and SFID scores.

| METHODS(↓), METRICS(→) | CLEVR | | TETROMINOES | | OBJECTS-ROOM | | BITMOJI | | FFHQ | |
|---|---|---|---|---|---|---|---|---|---|---|
| | FID(↓) | SFID(↓) | FID(↓) | SFID(↓) | FID(↓) | SFID(↓) | FID(↓) | SFID(↓) | FID(↓) | SFID(↓) |
| CoSA-GUMBEL | 116.1 | 257.7 | 32.1 | 256.1 | 194.1 | 184.2 | 127.1 | 262.6 | 225.7 | 237.1 |
| CoSA-EUCLIDIAN | 122.2 | 236.2 | 36.3 | 278.4 | 194.1 | 184.2 | 114.3 | 215.8 | 209.3 | 241.3 |
| CoSA-COSINE | 117.8 | 240.1 | 28.1 | 256.8 | 184.2 | 186.0 | 114.3 | 273.2 | 185.8 | 235.2 |

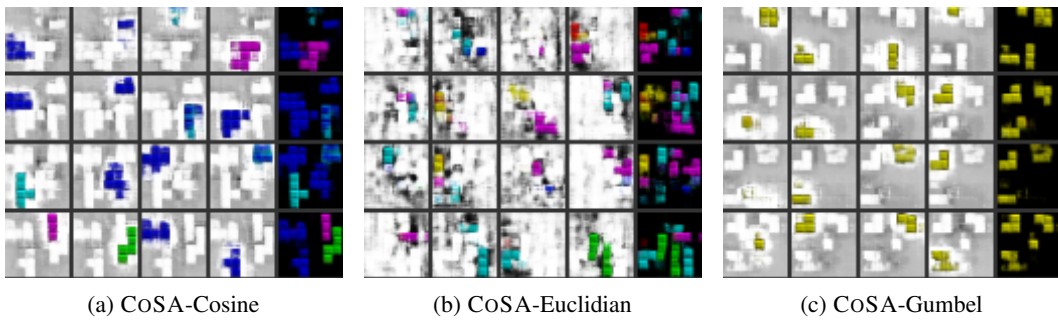

(a) CoSA-Cosine         (b) CoSA-Euclidian         (c) CoSA-Gumbel

Figure 26: Image composition with random slot prompt on CLEVR dataset.

To demonstrate the generalizability property of model due to grounded representations, we perform cross domain adaptability test. For adaptability test we train the model on one dataset and fine-tune the classification head on other datasets by controlling the data samples. The main hypothesis here is that the grounded representations are meaningful environmental properties, which goes beyond the downstream tasks. Qualitatively, table 16 describes the performance of the model trained and tested on CLEVR-Hans3 and CLEVR-Hans7 datasets in columns 1 and 3, while the other two columns indicate the task adaptability results with few shot (k=100) learning performances with the different base model. Similarly, table 15, 17 and table 18 describe the results on FloatingMNIST2 and FloatingMNIST3 datasets respectively with different variants of tasks (Add, Sub, and Mixed). Finally, table 19 demonstrates the performance on Mixed tasks on FloatingMNIST2 and FloatingMNIST3.

## M TRAINING AND COMPUTATIONAL DETAILS

We train all the models with adam optimizer with a learning rate of 0.0004, batch size of 16, with early stopping with the patience of 5, and for a maximum of min(40epochs, 40000 steps). We use linear learning rate warmup for 10000 steps with a decay rate of 0.5 followed by reduce on the plateau learning rate scheduler. The experimental config files and training/inference scripts with be made available for reproducibility and further research.

We run all our experiments on a cluster with Nvidia Telsa T4 16GB GPU card with Intel(R) Xeon(R) Gold 6230 CPU. The processing speed of our framework in comparison to the baselines is tabulated

(a) CoSA-Cosine         (b) CoSA-Euclidian         (c) CoSA-Gumbel

Figure 27: Image composition with random slot prompt on Tetrominoes dataset.

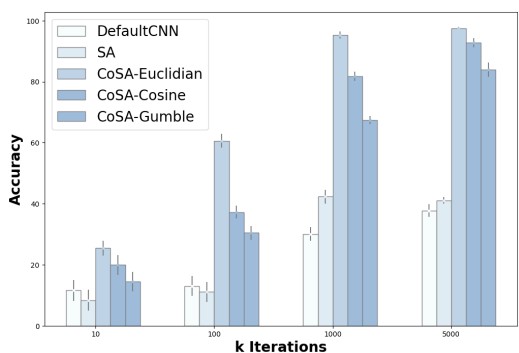

Figure 28: Few shot task adaptability accuracy for all the methods.

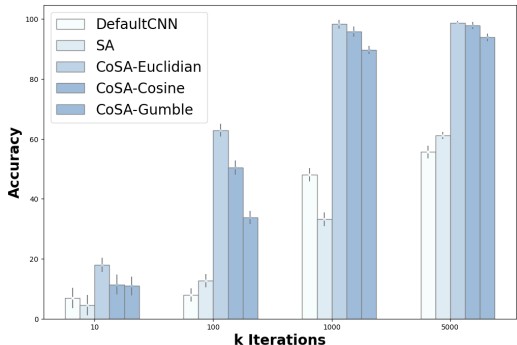

Figure 29: Few shot adaptability on FloatingMNIST2-diff to FloatingMNIST2-add dataset.

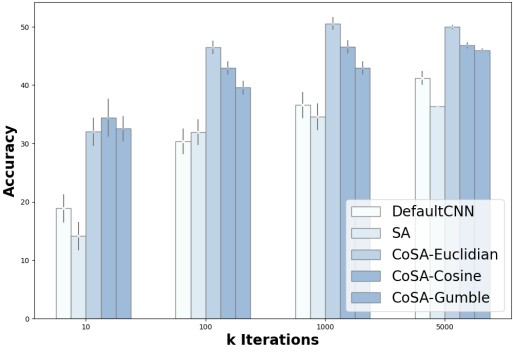

Figure 30: Few shot adaptability on CLEVR-hans3 to CLEVR-hans7 dataset.

Table 15: Accuracy and Hungarian matching coefficient (HMC) for reasoning objective on addition and subtraction variant of floatingMNIST2 (FMNIST2) dataset. Here, the first and third pair of columns correspond to the models trained and tested on the FMNIST2-Add and FMNIST2-Sub datasets, respectively, while the second and fourth pair correspond to few-shot(k=100) adaptability results across datasets.

| METHODS($\downarrow$), METRICS($\rightarrow$) | FMNIST2-ADD$_{source}$ | | FMNIST2-SUB$_{target}$ | | FMNIST2-SUB$_{source}$ | | FMNIST2-ADD$_{target}$ | |
|---|---|---|---|---|---|---|---|---|
| | ACC($\uparrow$) | HMC($\downarrow$) | ACC($\uparrow$) | F1($\uparrow$) | ACC($\uparrow$) | HMC($\downarrow$) | ACC($\uparrow$) | F1($\uparrow$) |
| CNN | 97.62 | - | 10.35 | 10.05 | 98.16 | - | 12.35 | 9.50 |
| SA | 97.33 | 0.14 | 11.06 | 09.40 | 97.41 | 0.13 | 08.28 | 7.83 |
| COSA-GUMBEL | 97.52 | 0.11 | 30.75 | 28.23 | 98.44 | 0.12 | 28.45 | 20.53 |
| COSA-EUCLIDIAN | **98.12** | **0.10** | **60.24** | **50.16** | **98.64** | 0.12 | **63.29** | **58.29** |
| COSA-COSINE | **98.33** | **0.10** | 37.47 | 36.00 | **98.81** | 0.12 | 50.43 | 44.47 |

Table 16: Accuracy and Hungarian matching coefficient (HMC) for reasoning objective on CLEVR-Hans dataset. Here, the first and third pair of columns correspond to the models trained and tested on CLEVR-Hans3 and CLEVR-Hans7 datasets respectively, while the second and fourth pair corresponds to few shot(k=100) adaptability results across models.

| METHODS($\downarrow$), METRICS($\rightarrow$) | CLEVR-HANS3$_{source}$ | | CLEVR-HANS7$_{target}$ | | CLEVR-HANS7$_{source}$ | | CLEVR-HANS3$_{target}$ | |
|---|---|---|---|---|---|---|---|---|
| | ACC($\uparrow$) | HMC($\downarrow$) | ACC($\uparrow$) | F1($\uparrow$) | ACC($\uparrow$) | HMC($\downarrow$) | ACC($\uparrow$) | F1($\uparrow$) |
| CNN | 67.89 | - | 28.14 | 16.66 | 80.06 | - | 58.49 | 55.11 |
| SA | 68.53 | 0.38 | 31.69 | 26.21 | 81.12 | 0.41 | 70.08 | 68.28 |
| COSA-GUMBEL | **71.88** | **0.26** | 40.42 | 38.18 | **82.72** | 0.32 | 69.98 | 69.81 |
| COSA-EUCLIDIAN | 70.64 | 0.28 | **46.45** | **46.88** | 81.17 | **0.30** | 71.35 | 68.33 |
| COSA-COSINE | 70.28 | 0.28 | 43.23 | 45.05 | **82.28** | 0.31 | 72.82 | 70.41 |

below. Table 20 and 21 describe the object discovery and reasoning speed in terms of iteration/sec. It is to be noted that speed might differ slightly with respect to the considered system and the background processes.

All experimental scripts and scripts to generate datasets is available on GitHub at https://github.com/koriavinash1/CoSA.

# N  LIMITATIONS AND FUTURE WORK

Our proposed method also possesses a few limitations; here we detail them and also propose a few possible ways to address them:

- Limited variation in slot-prompting: As the current prompting is solely based on dictionary size, the diversity in scene generation is limited. To address this, we can explore the possibility of stochastic VQVAE Takida et al. (2022). Additionally, contrastive learning could also prove useful to learn shape, colour, location, and texture codebooks separately, increasing the control and variability of compositional scene generation.

- Assumption of representation overcrowding: the assumption that representation overcrowding can be controlled with the architectural design, but exploring the direction to enforce this behaviour with additional regularization would be interesting.

- Heuristic for slot-distribution selection in the case of non-unique object composition: The current heuristic to handle repeated objects in a scene is dependent on the scale of principle components (eigenvalues of the principle direction vectors); it would be interesting to explore the directions to have it implicitly within the sampling procedure.

Table 17: Accuracy and Hungarian matching coefficient (HMC) for reasoning objective on addition and subtraction variant of floatingMNIST2 (FMNIST2) dataset. Here, the first and third pair of columns correspond to the models trained and tested on the FMNIST2-Add and FMNIST2-Sub variant of datasets respectively, while the second and fourth pair corresponds to k-shot adaptability results across models. Here, Base models corresponds to the models trained on source datasets and datasets described in the following row correspond to the target datasets, used in generalisability.

| BASE MODELS (→) | FMNIST2-MIXED | | FMNIST2-MIXED | | FMNIST2-MIXED | | FMNIST2-ADD | | FMNIST2-SUB | |
| METHODS(↓), METRICS(→) | FMNIST2-MIXED | | FMNIST2-ADD | | FMNIST2-SUB | | FMNIST2-MIXED | | FMNIST2-MIXED | |
| | Acc(↑) | HMC(↓) | Acc(↑) | F1(↑) | Acc(↑) | F1(↑) | Acc(↑) | F1(↑) | Acc(↑) | F1(↑) |
|---|---|---|---|---|---|---|---|---|---|---|
| CNN | **98.93** | - | 23.96 | 18.13 | 13.53 | 09.56 | 14.01 | 07.10 | 05.62 | 04.22 |
| SA | 98.12 | 0.11 | 15.32 | 11.25 | 20.02 | 15.49 | 13.13 | 09.55 | 14.13 | 09.57 |
| CoSA-GUMBEL | 98.28 | 0.11 | 37.86 | 34.45 | 41.04 | 34.75 | 29.30 | 24.04 | 32.57 | 24.72 |
| CoSA-EUCLIDIAN | 98.12 | 0.10 | 53.22 | 45.98 | 63.05 | 50.15 | **50.59** | **44.48** | 60.39 | **53.15** |
| CoSA-COSINE | **98.40** | **0.08** | **56.21** | **49.30** | **64.60** | **58.18** | 40.54 | 37.56 | 50.63 | 43.44 |

Table 18: Accuracy and Hungarian matching coefficient (HMC) for reasoning objective on addition and subtraction variant of floatingMNIST2 (FMNIST2) dataset. Here, the first and third pair of columns correspond to the models trained and tested on the FMNIST2-Add and FMNIST2-Sub variant of datasets respectively, while the second and fourth pair corresponds to k-shot adaptability results across models.

| BASE MODELS (→) | FMNIST3-MIXED | | FMNIST3-MIXED | | FMNIST3-MIXED | | FMNIST3-ADD | | FMNIST3-SUB | |
| METHODS(↓), METRICS(→) | FMNIST3-MIXED | | FMNIST3-ADD | | FMNIST3-SUB | | FMNIST3-MIXED | | FMNIST3-MIXED | |
| | Acc(↑) | HMC(↓) | Acc(↑) | F1(↑) | Acc(↑) | F1(↑) | Acc(↑) | F1(↑) | Acc(↑) | F1(↑) |
|---|---|---|---|---|---|---|---|---|---|---|
| CNN | 96.23 | - | 05.29 | 04.61 | 07.06 | 08.13 | 10.66 | 04.11 | 10.19 | 04.99 |
| SA | 96.29 | 0.18 | 08.83 | 04.77 | 06.68 | 03.40 | 07.16 | 03.87 | 13.81 | 06.28 |
| CoSA-GUMBEL | **97.37** | **0.14** | 14.13 | 08.53 | 21.77 | 12.31 | 11.95 | 09.31 | 18.95 | 12.28 |
| CoSA-EUCLIDIAN | 97.07 | 0.15 | 16.32 | 11.26 | 35.46 | 22.09 | **23.37** | **18.97** | 20.50 | **11.76** |
| CoSA-COSINE | 96.85 | 0.15 | **17.87** | **13.04** | 35.35 | **30.38** | 19.01 | 11.95 | **20.94** | 10.47 |

Table 19: Accuracy and Hungarian matching coefficient (HMC) for reasoning objective on addition and subtraction variant of floatingMNIST3 (FMNIST3) dataset. Here, the first and third pair of columns correspond to the models trained and tested on FMNIST3-Add and FMNIST3-Sub variant of datasets respectively, while the second and fourth pair corresponds to k-shot adaptability results across models.

| BASE MODELS (→) | FMNIST3-ADD | | FMNIST3-ADD | | FMNIST3-SUB | | FMNIST3-SUB | |
| METHODS(↓), METRICS(→) | FMNIST3-ADD | | FMNIST3-SUB | | FMNIST3-SUB | | FMNIST3-ADD | |
| | Acc(↑) | HMC(↓) | Acc(↑) | F1(↑) | Acc(↑) | HMC(↓) | Acc(↑) | F1(↑) |
|---|---|---|---|---|---|---|---|---|
| CNN | 96.42 | - | 15.92 | 08.10 | 97.62 | - | 06.80 | 04.46 |
| SA | 96.66 | 0.16 | 12.38 | 07.25 | 97.33 | 0.14 | 05.45 | 02.67 |
| CoSA-GUMBEL | **98.76** | 0.14 | 18.92 | 10.11 | 97.52 | 0.11 | 09.59 | 08.01 |
| CoSA-EUCLIDIAN | 98.11 | 0.14 | 19.82 | 11.72 | **98.12** | **0.10** | **17.91** | **11.03** |
| CoSA-COSINE | 98.33 | **0.10** | **20.22** | **13.63** | **98.33** | **0.10** | 14.01 | 07.82 |

Table 20: Iteration per second details for all datasets on object discovery task.

| Methods(↓), Metrics(→) | CLEVR | TETROMINOES | OBJECTS-ROOM | BITMOJI | FFHQ |
| | it/s | it/s | it/s | it/s | it/s |
|---|---|---|---|---|---|
| SA | 10 | 16 | 16 | 10 | 5 |
| IMPLICIT | 14 | 22 | 22 | 14 | 6 |
| CoSA-Gumbel | 7 | 18 | 18 | 7 | 4 |
| CoSA-Euclidian | 9 | 20 | 20 | 9 | 5 |
| CoSA-Cosine | 9 | 20 | 20 | 9 | 5 |

Table 21: Iteration per second details for all datasets on reasoning task

| METHODS(↓), METRICS(→) | CLEVR-HANS it/s | FLAOTINGMNIST it/s |
|---|---|---|
| SA | 16 | 16 |
| CNN | 20 | 20 |
| COSA-GUMBEL | 12 | 12 |
| COSA-EUCLIDIAN | 15 | 15 |
| COSA-COSINE | 15 | 15 |

