# OpenReview forum: "Grounded Object-Centric Learning"
_ICLR.cc/2024/Conference — ICLR 2024 poster_

### Official Review · Reviewer_myKR · 2023-10-26

**Soundness:** 3 good
**Presentation:** 3 good
**Contribution:** 3 good
**Rating:** 6
**Confidence:** 3

**Summary:**

The paper introduces Conditional Slot Attention (CoSA), a novel variant of Slot Attention that incorporates the concept of grounded representations. Unlike the original Slot Attention, CoSA utilizes a dynamic binding scheme using canonical object-level property vectors and parametric Gaussian distributions. This approach enables specialized slots that remain invariant to identity-preserving changes in object appearance. The proposed method is evaluated on multiple downstream tasks, including scene generation, composition, and task adaptation, and achieves competitive performance compared to Slot Attention in object discovery benchmarks.

**Strengths:**

1. Unsupervised object discovery remains a challenge and an open question in the research community.
2. The concept of the Grounded Slot Dictionary (GSD) module is logical, particularly the construction of a dictionary as outlined in Definition 1.
3. The visualization of GSD binding in Figure 3 is both interesting and insightful for the community, providing evidence of the effectiveness of the GSD approach.

**Weaknesses:**

1. I would appreciate more ablation studies in the experiments section. The current version primarily presents the state-of-the-art (SOTA) performance for two case studies, but additional ablation studies would provide further insights into the specific contributions and the impact of different components or techniques employed in the proposed method.
2. The author mentions that the method incorporates the object-level property vector, but there is a lack of evidence regarding how it functions. For instance, it is unclear whether the method can effectively discriminate between multiple instances with similar appearances.
3. The visualization of the COCO results does not appear to be accurate, and it seems that the method may not be as applicable to real-world scenarios.

**Questions:**

1. Besides performance gains, what evidence does the paper provide to show that the object-level property vectors are working effectively?

---

> ### Author Response · Authors · 2023-11-19
>
> We thank the reviewer for their detailed comments and constructive feedback. We very much appreciate the fact that our framework was received as logical, interesting, and insightful for the community.
>
> > I would appreciate more ablation studies in the experiments section. The current version primarily presents the state-of-the-art (SOTA) performance for two case studies, but additional ablation studies would provide further insights into the specific contributions and the impact of different components or techniques employed in the proposed method.
>
> We agree that the ablation can provide more insights about the method; we have included ablations on sampling, types of spectral decomposition, and sample efficiency on multiple tasks and datasets. Additionally, we have included the ablations on codebook size and its implications on results in the appendix.
> In case we are missing any ablations that would add to the clarity in the context of CoSA, please let us know.
>
>
> > The author mentions that the method incorporates the object-level property vector, but there is a lack of evidence regarding how it functions. For instance, it is unclear whether the method can effectively discriminate between multiple instances with similar appearances
>
> The evidence of selecting object-level property vectors can be observed in both object discovery and reasoning tasks. In the case of object discovery, FG-ARI measures this property, whereas HMC measures the same for reasoning tasks measures this property, whereas for reasoning tasks, HMC measures the same.
>
> Similarly, in the case of Object discovery, we have added new results to qualitatively demonstrate the binding across multiple instances; please refer to Figure 19 in the appendix section.
>
>
> > The visualization of the COCO results does not appear to be accurate, and it seems that the method may not be as applicable to real-world scenarios.
>
>  We do observe that the results on the COCO dataset are not as good as supervised segmentation models, but it is important to note that the proposed model is fully unsupervised, and based on the initial number of slots hyperparameter, the model does segregate between different types of objects in the image. We leave further exploration in improving the model performance to match closely to the segmentation ground truths to future work.
>
>
> > Besides performance gains, what evidence does the paper provide to show that the object-level property vectors are working effectively?
>
> We demonstrate the notion of binding and qualitatively as well as quantitively. Qualitatively, this can be seen in the grounded dictionary visualization section in the main paper (Figure 3) and in appendix section K.1 and section H.5. Quantitively, the results are reflected in terms of FG-ARI and HMC metrics.

---

> > ### Comment · Reviewer_myKR · 2023-11-20
> >
> > Thank you for the detailed response. It addresses most of my concerns. I believe this paper presents significant insights and contributions to slot attention and object discovery. Thus, I will maintain my current rating.

---

> > > ### Author Response · Authors · 2023-11-20
> > >
> > > Many thanks for the swift reply. We are very glad to see the reviewer is happy with our revision and sees significant insights and contributions in our work.
> > >
> > > Given the very positive feedback to our revision, we would kindly ask the reviewer to consider updating their score to better reflect their feedback. Otherwise, please let us know if there are any remaining concerns we can address.
> > >
> > > Many thanks again for taking the time to assess our work. This is much appreciated.

---

### Official Review · Reviewer_yF2M · 2023-10-30

**Soundness:** 3 good
**Presentation:** 3 good
**Contribution:** 3 good
**Rating:** 6
**Confidence:** 4

**Summary:**

This paper focuses on grounded object-centric learning which can bind to specific object types. To achieve this goal, the authors take inspiration from Slot Attention, and introduce a Grounded Slot Dictionary to encode object properties and bind to different object types. This dictionary enables the model to conditionally sample the slots from different distributions. And the reasoning module with property transformation module enhances the interpretability of object property binding. The experiments show improvements on various object discovery benchmarks.

**Strengths:**

1. The motivation is clear and makes sense. It takes inspiration from the recent research in binding problem and concludes three major challenges in unsupervised object discovery. And it takes the binding to canonical object properties and types as the primary problem to help simultaneously solve all three challenges.
2. The design of conditional slot attention is novel. It employs the spectral decomposition for discrete mapping and enables the model to sample from different distributions corresponding to different object properties.
3. The visualization of the separated object properties are interesting and shows interpretable Ground Slot Dictionary.

**Weaknesses:**

1. The experiments on more complex scenes are required. For example, multiple instances of the same object category, it would be interesting to show the property binding ability in this case.
2. Does the model build on pre-trained backbones? Or a randomly initialized encoder $\Phi$ is also sufficient to provide cues for spectral decomposition and discritization?
3. I suggest authors to run the trained conditional slot attention with grounded slot dictionary on some video data, e.g., DAVIS-2017, to validate whether the slot dictionary can track objects across time and more vividly show the binding ability to object types.

**Questions:**

The conventional slot attention based methods use reconstruction loss as the self-supervised objective to guide object decomposition. What is the relationship between the objectives used in this work and the recounstruction loss?

---

> ### Author Response · Authors · 2023-11-19
>
> We thank the reviewer for their detailed comments and constructive feedback. We very much appreciate the fact that our work was found to be well-motivated, novel, and the illustrations were found interesting and interpretable.
>
> > The experiments on more complex scenes are required. For example, multiple instances of the same object category, it would be interesting to show the property binding ability in this case.
>
> We share the reviewers view on examining the binding ability of the framework on multiple object instances of the same object. Given that CLEVR has several of these examples in the dataset, we included a section in Appendix H.5, where we demonstrate the reusability of dictionary elements based on repeated object instances. As illustrated in Figure 19, similar looking object slot representations are sampled from the same distribution.
>
>
> > Does the model build on pre-trained backbones? Or a randomly initialized encoder is also sufficient to provide cues for spectral decomposition and discritization?
>
> For CLEVR, ObjectsRoom, Bitmoji, FFHQ, and Tetrominoes datasets we start with a randomly initialized encoder.
> For COCO, given the very high diversity of image content, we start with a pre-trained image encoder similar to DINOSAUR and apply CoSA on the extracted features.
>
>
> > I suggest authors to run the trained conditional slot attention with grounded slot dictionary on some video data, e.g., DAVIS-2017, to validate whether the slot dictionary can track objects across time and more vividly show the binding ability to object types.
>
> We thank the reviewer for the great suggestion. We agree that extending the proposed method to a large-scale video dataset like DAVIS-2017 would be interesting and valuable. However, at this stage, we believe that this would be best served by a dedicated investigation in future work.
>
> Nonetheless, to reassure the review of our method’s ability to correctly identify and bind multiple object instances, we have added a discussion section and some illustrations using  CLEVR in the appendix (section H.5).
>
>
>
> > The conventional slot attention based methods use reconstruction loss as the self-supervised objective to guide object decomposition. What is the relationship between the objectives used in this work and the reconstruction loss?
>
> The objective function in our work can be viewed as the sum of reconstruction loss (likelihood loss) + KL regularization. The additional KL term is used to train the strategies of selecting the elements from GSD for a given image.
>
> Our work introduces a probabilistic perspective of slot attention by optimizing a principled variational lower bound of the log likelihood of the data. The image likelihood we maximize is related to the reconstruction loss in previous works in the sense that minimzing a MSE loss is analogous to maximizing a pixel-wise factored Gaussian likelihood.

---

> > ### Comment · Reviewer_yF2M · 2023-11-21
> >
> > Thanks for the response. However, I am more interested in the model's ability to discriminate different instances of same semantics in realistic scenes, instead of the synthetic CLEVR dataset. From Fig.19, it is clear that it is feasible to discriminate objects solely by the color and shape or size, which is much simpler than the real world applications. Since DAVIS-2017 contains some videos that consist of multiple instances of same semantics, e.g., dogs-jump, goldfish, soapbox, I kindly request the authors to just run inference without retraining on such video sequences and show the visualization on consecutive frames.

---

> ### Author Response · Authors · 2023-11-21
>
> Thank you for your response and feedback.
>
> Based on your suggestion, we have included inference results on a video from the DAVIS dataset. Please refer to Figure 20 in the appendix, even in this case, we can observe that the CoSA consistently binds foreground and background slots to a particular GSD element.

---

> > ### Comment · Reviewer_yF2M · 2023-11-22
> >
> > After reading the resutls in the appendix, I agree with the author's claim that CoSA consistenty binds foreground and background slots to a particular GSD element. However, the author only shows the results on simple one foregroud (a camel) vs background, this foreground background separation and binding attribute has been verified in [1] with very simple query slot attention. Simply replacing the slot initialization with learnable queries equips the model with this binding ability. I still insist that the authors should validate the binidng attribute of in more complex video sequences with multiple objects to show the advantage of the proposed framework over the simple query slot attention. It is important to show whether the learned model consistency bind specific semantics or instances in a video sequence to particular GSD elements.
> >
> > [1] Jia, B., Liu, Y., & Huang, S. Improving object-centric learning with query optimization. ICLR 2023.

---

> > > ### Author Response · Authors · 2023-11-22
> > >
> > > We thank the reviewer for taking the time to engage with us and for their constructive feedback.
> > > We also appreciate the pointer towards the query optimization paper to our attention, we will add a reference to it in the final manuscript.
> > >
> > > We are pleased that the additional video results we added to the appendix helped convince the reviewer that our method can bind semantics to the same GSD element across time as requested.  In fact, it did so without explicit retraining on the new dataset, which we believe is interesting in its own right.
> > >
> > > Regarding the request for additional results in even more complex videos with multiple objects, we feel this would be best served by a dedicated investigation in future work. It is also worth noting that the transformer variant of the DINO backbone we used in our experiments on real-datasets has an inherent bias towards grouping multiple instances of the same object together (as observed in previous work see Figure 7 in [1]). This is a bias we would have to overcome in future work.
> > >
> > > Although we agree with the general sentiment of the reviewer in the sense that evaluating on more complex datasets always strengthens a submission, we believe our work makes multiple timely and significant contributions to object-centric learning as it stands. To reiterate, our main contributions are the following:
> > >
> > > - Our proposed Grounded Slot Dictionary (GSD) unlocks the capability of learning specialized slots that bind to specific object _types,_ as prescribed by classical notions of the grounding problem (Hardnad, 1990; Treissman 1999);
> > > - We provided a principled probabilistic perspective on unsupervised learning dictionary learning, and derive a principled end-to-end variational objective for this model class;
> > > - We introduce a simple but effective strategy for dynamically estimating the number of unique and repeated objects in a given scene, enabling dynamic slot estimation for the first time.
> > >
> > > [1] Seitzer et.al., Bridging the gap to real-world object-centric learning, ICLR 2023.

---

### Official Review · Reviewer_dFhu · 2023-11-01

**Soundness:** 2 fair
**Presentation:** 2 fair
**Contribution:** 3 good
**Rating:** 6
**Confidence:** 3

**Summary:**

This paper introduces Conditional Slot Attention (CoSA) that uses a Grounded Slot Dictionary (GSD) to sample initial slots from a shared library of canonical object-level property vectors. Spectral decomposition is used to estimate the number of initial slots $K$ and vector quantization is used to select initial slot distributions from the GSD. The authors run experiments on object discovery, scene composition and generation, and downstream task adaptation showing benefits over previous methods.

**Strengths:**

This paper is well-motivated and the approach is novel, as far as I know. The authors show improvements over previous methods in terms of FG-ARI and downstream task performance and include additional experiments and analyses in the Appendix.

**Weaknesses:**

- The paper is missing some ablations that I think would be important in evaluating the significance of the method:
    - How much of the improved performance is from just predicting the number of slots instead of using a fixed $K$? What if we use the predicted number of slots without the GSD?
    - Conversely, how much is attributed to the GSD? What if we used a fixed $K$ with GSD?
- What is the distribution of the number of objects for the different datasets? This would be important to interpreting the MAE values.
- While I can appreciate the probabilistic interpretation of the model, I feel it does not add to the clarity of the paper and may be more appropriate for the appendix. Specifically, if I understand correctly, if $q(\tilde{z}|x)$ is deterministic and $p(\tilde{z})$ is uniform as in VQ-VAE, then the KL term is just a constant and not actually used to optimize the model?
- The discussion of the different sampling strategies (Euclidean, Cosine, Gumbel) does not seem necessary for the main text since (from my understanding) the experiments in the main text are only done with the Cosine version? I do see additional experiments on other sampling strategies in the appendix, but if that is the case, this discussion can be removed from the main text.

**Questions:**

- The codebook size seems like a potentially important parameter. What size do you choose for the different experiments? How sensitive are the results to codebook size?
- I want to confirm that the Abstraction Module is not differentiable and just uses the output of the encoder, which is trained through the Slot Attention path. Then, there are similarly no gradients flowing through the GSD and it is only updated with EMA (Appendix F). Is this understanding correct? If so, I think this could be stated more explicitly for clarity in the main text.
- How are the dynamic number of slots $K$ actually implemented during training? From my understanding, different images in a batch may have different $K$, so this may need to be done with some masking of the softmax in Slot Attention, in which case a max number of slots still needs to be used. In that case, is the benefit in FLOPS only during inference for a single image?

---

> ### Author Response · Authors · 2023-11-19
>
> We thank the reviewer for their detailed comments and constructive feedback. We very much appreciate the generally positive outlook and the fact that our work was found to be well-motivated and novel.
>
> > How much of the improved performance is from just predicting the number of slots instead of using a fixed K? What if we use the predicted number of slots without the GSD?
>
> Our apologies for the confusion. Although it is technically possible to perform dynamic estimation of the number of slots K at training time using our method, it requires masking of surplus slots, which somewhat complicates the training pipeline without providing guaranteed benefits. Our method was used at inference time only; we have made this clear in the revised version of the paper.
>
> Therefore, comparing performance improvements from training with the dynamic estimation of K versus the GSD is not applicable, since the GSD was always trained with fixed K. If we remove the GSD, we end up with the standard Slot-Attention model, which we compare with in our ablation study (e.g. see the SA model in Table 1, Table 2 etc).
>
>
> > What is the distribution of the number of objects for the different datasets? This would be important to interpreting the MAE values.
>
> Thanks for highlighting this point, we have updated the dataset information in appendix section C, with the distribution of the number of objects for the different datasets.
>
>
> > While I can appreciate the probabilistic interpretation of the model, I feel it does not add to the clarity of the paper and may be more appropriate for the appendix. Specifically, if I understand correctly, if q(z|x) is deterministic and p(z) is uniform as in VQ-VAE, then the KL term is just a constant and not actually used to optimize the model?
>
> We thank the reviewer for the astute observation. It is true that under a uniform prior and deterministic sampling, the KL term drops to a constant. However, this is not the case for the Gumbel sampling scenario, as while the prior is still uniform, the sampling is stochastic. For this reason, the probabilistic exposition in our paper is designed to be generally consistent with any (discrete) representation learning setup.
>
> We have clarified this in the paper and have included additional background on quantization clarifying this in the appendix.
>
>
>
> > The discussion of the different sampling strategies (Euclidean, Cosine, Gumbel) does not seem necessary for the main text since (from my understanding) the experiments in the main text are only done with the Cosine version?
>
> We thank the reviewer for this suggestion; we have briefly introduced different types of sampling for the sake of completion, providing different options available for constructing the dictionary. We have included details of quantization and the sampling methods in the appendix.
>
>
> > The codebook size seems like a potentially important parameter. What size do you choose for the different experiments? How sensitive are the results to codebook size?
>
> In all our experiments, we initialize the codebook with a sufficiently large number of embeddings for the complexity of the datasets used. We’ve included additional ablations and further discussion on codebook size in the appendix section H.4.
>
> Here are the ablation results on the CLEVR dataset.
>
> | M| MSE | FG-ARI | CBD | CBP |
> | :----: |:----: |:----: |:----: |:----: |
> | | | Cosine | | |
> | M = 16 | 8.11  | 0.73 | 0.99 | 14.96 |
> | M = 64 | 3.14 | 0.96 | 0.99 | 44.39 |
> | M = 128 | 3.46  | 0.94 | 0.98 | 51.65 |
> | M = 256 | 6.68  | 0.85 | 0.98 | 82.20 |
> | | | Euclidian | | |
> | M = 16 | 4.81 | 0.93 | 0.99 | 8.33 |
> | M = 64 | 4.04 | 0.94 | 0.99 | 32.40 |
> | M = 128 | 3.64 | 0.94 | 0.95 | 20.69 |
> | M = 256 | 4.34 | 0.89 | 0.96 | 19.79 |
> | | | Gumble | | |
> | M = 16 | 6.64 | 0.88 | 1.0 | 11.06 |
> | M = 64 | 3.82 | 0.93 | 1.0 | 29.62 |
> | M = 128 | 3.89 | 0.93 | 0.99 | 27.16 |
> | M = 256 | 5.46 | 0.92 | 0.99 | 30.28 |
>
>
> > I want to confirm that the Abstraction Module is not differentiable and just uses the output of the encoder, which is trained through the Slot Attention path. Then, there are similarly no gradients flowing through the GSD and it is only updated with EMA (Appendix F). Is this understanding correct?
>
> Yes, the reviewer is indeed correct; the abstraction module is nonparametric, but the gradients flow through it.
>
> In the case of GSD, the key part of the dictionary $\mathfrak{S}^1$ is updated via EMA, while the distribution parameters in $\mathfrak{S}^2$ are updated with gradients from the SA module.
>
>
> > How are the dynamic number of slots actually implemented during training?
>
> Dynamic slot selection during training requires masking of surplus slots, which complicates the training pipeline without providing guaranteed benefits. For this reason, we trained with a fixed number of slots K, and apply dynamic slot selection at inference time only.

---

> > ### Author Response · Authors · 2023-11-22
> >
> > Dear Reviewer dFhu,
> > We appreciate your time and effort in reviewing our work. Given the time constraint, would you please check our responses and confirm, if all your concerns are addressed?
> > Thank you,  Authors

---

> > ### Comment · Reviewer_dFhu · 2023-11-22
> > **Response to Rebuttal**
> >
> > Thank you for taking the time to respond to my concerns and run additional ablation experiments. Given the response which addresses most of my questions and also reading the other reviews and responses, I've decided to increase my score to 6. However, I still do encourage the authors to simplify the main text by focusing on the Cosine variant which seems to be the one used in the main experiments. I feel like investigating different sampling strategies can almost form another paper and does not help in the understanding of the proposed method.
> >
> > I also could not find the updated text mentioning that estimating K is only done at inference time, although I may have missed it.
> >
> > Minor typo (that did not affect my decision): Gumble should be Gumbel

---

### Author Response · Authors · 2023-11-20

We thank all the reviewers for their time and effort in reviewing our work. We are glad that all the reviewers found our work well-motivated and novel, we also appreciate the fact that our contributions were perceived as ‘logical’, ‘interesting’, and ‘insightful for the community’.

To reiterate the main contributions in our work:

- We propose the Grounded Slot Dictionary for object-centric representation learning, which unlocks the capability of learning specialized slots that bind to specific object types. Notably, this differs from the standard SA, which relies on a single distribution for random slot initialization and does not encourage slots to remain invariant in the face of identity-preserving changes in object appearance;

- We provide a probabilistic perspective on unsupervised object-centric dictionary learning and derive a principled end-to-end objective for this model class using variational inference methods;

- We introduce a simple strategy for dynamically quantifying the number of unique and repeated objects in a given input scene by leveraging spectral decomposition techniques;

- We experimentally demonstrate the benefits of grounded slot representations in multiple tasks such as scene generation, composition and task adaptation whilst remaining competitive with standard slot attention in object discovery-based tasks.



We took all the reviewers' comments into account and made the following main changes (highlighted in blue in the revised manuscript):

- Detailed description on quantization, and how KL term in the object affects the learning based on the choice of sampling (Appendix F);

- Ablation on codebook size: we included an extensive ablation on codebook size for all three sampling strategies based on the CLEVR dataset. We mainly observe that the proposed framework is not extremely sensitive to the codebook size, given the codebook does not collapse; we additionally include strategies to prevent codebook collapse (Appendix H.4 and Appendix G);

- We have updated the dataset information to include the maximum object distribution per image in all object discovery datasets (tabulated in Table 5);

- In addition, we included a section demonstrating binding on a similar-looking object instances in a given image. We mainly show that similar looking object instances in an environment sample the same GSD element, highlighting the binding property (demonstrated in Figure 19).

---

### Meta-Review · Area_Chair_L581 · 2023-12-11

**Metareview:**

This paper proposes to learn multiple slot priors designated to specific objects instead of a single prior. To this end, the authors introduced conditional Slot Attention, which maintains the parameters for multiple priors similar to codebooks in vector quantization.

The paper received borderline scores from the reviewers. The primary concerns from the reviewers were about insufficient ablation studies and missing comparisons on realistic datasets. The authors adequately addressed some of these concerns in the rebuttal, and the reviewers all recommended borderline acceptance.

After reading the paper, reviews, and rebuttal, the AC agrees with the reviewers' decision and recommends acceptance. The authors should include the additional results and clarifications presented in the rebuttal to the camera-ready version of the paper.

**Justification For Why Not Higher Score:**

The idea of learning multiple priors, or binding the objects to specific slots, has been already investigated in prior works. The proposed paper presents a reasonable approach along this line, but both the technical contributions and the significance of the results are not particularly strong to be highlighted at the conference.

**Justification For Why Not Lower Score:**

N/A

---

### Decision · Program_Chairs · 2024-01-16

Accept (poster)